



# Acceleration of Tropical Cyclones As a Proxy For Extratropical Interactions: Synoptic-Scale Patterns and Long-Term Trends

Anantha Aiyyer[1] and Terrell Wade[1]

[1]Department of Marine, Earth and Atmospheric Sciences, North Carolina State University

**Correspondence:** A. Aiyyer (aaiyyer@ncsu.edu)

**Abstract.** It is well known that rapid changes in tropical cyclone motion occur during interaction with extratropical waves. While the translation speed has received much attention in the published literature, acceleration has not. Using a large data sample of Atlantic tropical cyclones, we formally examine the composite synoptic-scale patterns associated with *tangential* and *curvature* components of their acceleration. During periods of rapid tangential acceleration, the composite tropical cy-

clone moves poleward between an upstream trough and downstream ridge of a developing extratropical wavepacket. The two systems subsequently merge in a manner that is consistent with extratropical transition. During rapid curvature acceleration, a prominent downstream ridge promotes recurvature of the tropical cyclone. In contrast, during rapid tangential or curvature deceleration, a ridge is located directly poleward of the tropical cyclone. Locally, this arrangement takes the form of a cyclone-anticyclone vortex pair somewhat akin to a dipole block. On average, the tangential acceleration peaks 18 hours prior

to extratropical transition while the curvature acceleration peaks at recurvature. These findings confirm that rapid acceleration of tropical cyclones is mediated by interaction with extratropical baroclinic waves. Furthermore, The tails of the distribution of acceleration and translation speed show a robust reduction over the past 5 decades. We speculate that these trends may reflect the poleward shift and weakening of extratropical Rossby waves.

## 1 Introduction

The track and movement of tropical cyclones are known to be governed by the background environment (e.g., Hodanish and Gray, 1993). It was recognized early on that the translation speed of a tropical cyclone can be approximated by the surrounding wind field (Emanuel, 2018). A tropical cyclone, however, is not an isolated vortex that is passively carried by the current. The background environment is comprised of synoptic and large-scale circulation features, with attendant gradients of potential vorticity, moisture, and deformation. The tropical cyclone actively responds to these external stimuli. The large

scale environment is also impacted by the interaction. For example, the generation of $\beta$-gyres – that influence tropical cyclone motion – is a response to the background potential vorticity gradient (e.g., Chan and Williams, 1987; Wu and Emanuel, 1993). Similarly, vertical wind shear can limit the intensification of tropical cyclones by importing low entropy air (e.g., Tang and Emanuel, 2010). This, in turn, can impact subsequent tropical cyclone motion. On the other hand, a tropical cyclone can influence the large-scale flow by exciting waves in the extratropical stormtrack, leading to rapid downstream development and



rearrangement of the flow (e.g., Jones et al., 2003). It can be contended that a tropical cyclone is always interacting with its environment, and the interaction is partly manifested in its motion.

Track and translation speed are two aspects of tropical cyclone motion that are particularly important for operational forecasts. The track garners much attention for obvious reasons – it informs potential locations that will be affected by a storm. The translation speed impacts intensity change, storm surge and local precipitation amount. There is a large body of published

literature and review articles dealing with various research and operational problems related to tropical cyclone track and speed (e.g., Chan, 2005; Emanuel, 2018). When tropical cyclones move poleward, they often encroach upon the extratropical storm-track. This leads to their interaction with baroclinic eddies of the stormtrack. The outcome of the interaction is varied. Some tropical cyclones weaken and dissipate while others strengthen and retain their tropical nature for an additional period. In the North Atlantic, around 50% of tropical cyclones eventually experience extratropical transition - a complex process during

which the warm-core tropical cyclone evolves into a extratropical cyclone and becomes part of an extratropical stormtrack (e.g., Hart and Evans, 2001). Several recent reviews have extensively documented the research history and dynamics of extratropical transition (e.g., Evans et al., 2017; Keller et al., 2019). Forecasters have long known that tropical cyclones accelerate forward during extratropical transition, but relatively less attention has been devoted to the details of this acceleration in the research literature.

This paper has two main themes. The first concerns the synoptic-scale flow associated with rapid acceleration and deceleration of tropical cyclones. This is addressed via composites of global reanalysis fields and observed tropical cyclone tracks from 1980–2016. We consider both tangential and curvature accelerations. To our knowledge, such formal delineation of extratropical interaction based on categories of tropical cyclone acceleration has not been presented in the literature. Of note, however, is the recent work of Riboldi et al. (2019) that is relevant to this paper. That study examined the interaction of accelerating

and decelerating upper-level troughs and recurving western North Pacific typhoons. Their key findings are: (a) In the majority of cases, a recurving tropical cyclone is associated with a decelerating upper-level trough that remains upstream; (b) The upper-level trough appears to phase lock with the tropical cyclone; and (c) Recurvatures featuring such trough deceleration are frequently associated with downstream atmospheric blocking. As we shown in subsequent sections, many of these results can be recovered independently when we approach the problem from the perspective of tropical cyclone motion. As such, part of

our work complements the findings of Riboldi et al. (2019).

The second theme concerns the identification of long-term trends in tropical cyclone motion. While we focus on acceleration, we begin with translation speed to place our results within the context of related recent work. Kossin (2018) reported that the translation speed of tropical cyclones over the period 1949–2016 has reduced by about 10% over the globe, and about 6% over the North Atlantic. Without directly attributing it, Kossin (2018) noted that the trend is consistent with the observed

slow-down of the atmospheric circulation due to anthropogenic climate change. In addition to the circulation changes, the general warming of the atmosphere is associated with an increase in water vapor content per the Clausius-Clapeyron scaling (CCS). This translates to increase in precipitation rates that can locally exceed the CCS (e.g., Nie et al., 2018). Kossin (2018) made the point that the reduced translation speed may compound the problem of heavy precipitation in a warmer climate. However, Moon et al. (2019) and Lanzante (2019) argued that the historical record of tropical cyclone track data, particularly





prior the advent of the satellite era around the mid-1960s, is likely incomplete. They also showed that annual-mean tropical cyclone translation speed exhibits step-like changes and questioned the existence of a true monotonic trend. They attributed these discrete changes natural factors (e.g., regional climate variability) as well as omissions and errors due to lack of direct observation prior to the availability of extensive remote sensing tools.

We revisit the issue of trends in tropical cyclone motion, but restrict our attention to the Atlantic basin and the years since
1966. This year is considered to be the beginning of the *satellite era*, at least for the Atlantic (Landsea, 2007). In contrast, global coverage by geostationary satellites began much later, in 1981 (e.g., Schreck et al., 2014). Lanzante (2019) found a change point in 1965 for the timeseries of annual average tropical cyclone speed for the North Atlantic basin. Lanzante (2019) also reported that accounting for the change point dramatically reduces the trend from the value reported by Kossin (2018). In light of the issues raised about the reliability of tropical cyclone track data prior to the *satellite era*, we use 1966 as our starting
point and seek to ascertain whether robust trends exist in the observed record of tropical cyclone acceleration.

## 2   Data

We use the IBTrACS v4 (Knapp et al., 2010) for tropical cyclone locations. The information in this database typically spans the tropical depression stage to extratropical cyclone and/or dissipation. Kossin (2018), used all locations in the IBTrACs as long as a storm lasted more than 3 days and did not apply a threshold for maximum sustained winds. We follow the same
method with one major difference. We only consider those instances of the track information wherein a given storm was still classified as tropical. We recognize the fact that the nature of a storm is fundamentally altered after it loses its surface enthalpy flux-driven, warm-core structure. After extratropical transition, evolution is governed by stormtrack dynamics of baroclinic eddies. We wish to avoid conflating the motions of two dynamically distinct weather phenomena. For the same reason, we also omit subtropical storms which are catalogued in the track database. We argue that, by restricting our track data to only
instances that were deemed to be tropical in nature, we can paint a more appropriate picture of the composite environment and trends associated with the rapid acceleration of tropical cyclones.

To ascertain whether a tropical storm underwent extratropical transition, we use the *nature* designation in the IBTrACS database that relies on the judgement of the forecasters from one or more agencies responsible for an ocean basin. In IBTrACS, the nature flag is set to "ET" after the transition is complete. Admittedly, there will be some subjectivity in this designation. An
alternative would be to employ a metric such as the cyclone phase space (Hart, 2003), usually calculated using meteorological fields from global numerical weather prediction models (e.g., reanalysis). As we wish to remain independent of modeled products to characterize the storms, we rely on the forecaster designated storm nature. Furthermore, Bieli et al. (2019) found that the phase space calculation was sensitive to the choice of the reanalysis model, which would add another source of uncertainty. Occasionally, the nature of the storm is not recorded (NR) or, if there is an inconsistency among the agencies, it
is designated as mixed (MX). In the north Atlantic, only a small fraction of track data ($\approx 0.5\%$) in the IBTrACs is designated as NR or MX. This fraction is higher in other basins (e.g., $\approx 14\%$ in the western North Pacific). This is another reason why





we restrict our attention to the Atlantic basin. Henceforth, we will collectively refer to the designations NR, MX and ET as *non-tropical*, and unless explicitly stated, omit the associated track data in our calculations.

For the trends and basic statistics, we focus on the years 1966–2019, within the ongoing *satellite era* for the North Atlantic
basin (e.g. Landsea, 2007; Vecchi and Knutson, 2008). Compared to years prior, tropical cyclone data is deemed more reliable once satellite-based observations became available. For the composites, we use the European Center for Medium Range Weather Forecasting (ECMWF) ERA-Interim (ERAi) reanalysis (Dee et al., 2011) for the period 1981–2016. In subsequent sections, we focus on the region 20–50$^o$N, wherein tropical cyclones are more likely to interact with extratropical baroclinic waves.

## 3  Tangential and curvature acceleration

The acceleration of a hypothetical fluid element moving with the center of a tropical cyclone can be written as:

$$\mathbf{a} = \frac{dV}{dt}\hat{\mathbf{s}} + \frac{V^2}{R}\hat{\mathbf{n}} \tag{1}$$

where $V$ is the forward speed and $R$ is the radius of curvature of the track at a given location. Here, $\hat{\mathbf{s}}$ and $\hat{\mathbf{n}}$ are orthogonal unit vectors in the so-called *natural coordinate* system. The former is directed along the tropical cyclone motion. The latter is
directed along the radius of curvature of the track. The first term in eq. 1 is the tangential acceleration and the second is the curvature or normal acceleration. The speed $V_j$ at any track location (given by index j) is calculated as:

$$V_j = \frac{(D_{j,j-1} + D_{j,j+1})}{\delta t} \tag{2}$$

where $D$ refers to the distance between the two consecutive points, indexed as shown above, along the track. Since we used 3-hourly reports from the IBTrACS, $\delta t = 6$ h. The tangential acceleration was calculated from the speed using centered differ-
ences.

To calculate the curvature acceleration, it is necessary to first determine the radius of curvature, R. A standard approach to calculating R, given a set of discrete points along a curve – in our case, a tropical cyclone track – is to fit a circle through three consecutive points. For a curved line on a sphere, it can be shown that:

$$R = R_e sin^{-1}\left(\sqrt{\frac{2d_{12}d_{13}d_{23}}{(d_{12}+d_{13}+d_{23})^2 - 2(d_{12}^2 + d_{13}^2 + d_{23}^2)}}\right) \tag{3.1}$$

where $R$ is the radius of curvature, $R_e$ is the radius of the Earth, and the $d$ terms are expressed as follows:

$$d_{12} = 1 - (\cos T_1 \cos T_2 \cos (N_2 - N_1) + \sin T_1 \sin T_2) \tag{3.2}$$

$$d_{13} = 1 - (\cos T_1 \cos T_3 \cos (N_3 - N_1) + \sin T_1 \sin T_3) \tag{3.3}$$

$$d_{23} = 1 - (\cos T_2 \cos T_3 \cos (N_3 - N_2) + \sin T_2 \sin T_3) \tag{3.4}$$



where, $T_1$, $T_2$, and $T_3$ are the latitudes of the 3 points while $N_1$, $N_2$, and $N_3$ are the longitudes. The center of the circle is given by the coordinates:

$$\tan\mathbf{T} = \pm\frac{\cos T_1\cos T_2\sin(N_2-N_1)+\cos T_1\cos T_3\sin(N_1-N_3)+\cos T_2\cos T_3\sin(N_3-N_2)}{\sqrt{\alpha^2+\beta^2}} \tag{4.1}$$

$$\tan\mathbf{N} = -\frac{\alpha}{\beta} \tag{4.2}$$

Where $\mathbf{T}$ and $\mathbf{N}$ are the latitude and longitude, respectively of the circle's center, and $\alpha$ and $\beta$ are obtained using:

$$\alpha = \cos T_1(\sin T_2-\sin T_3)\cos N_1+\cos T_2(\sin T_3-\sin T_1)\cos N_2+\cos T_3(\sin T_1-\sin T_2)\cos N_3 \tag{4.3}$$

$$\beta = \cos T_1(\sin T_2-\sin T_3)\sin N_1+\cos T_2(\sin T_3-\sin T_1)\sin N_2+\cos T_3(\sin T_1-\sin T_2)\sin N_3 \tag{4.4}$$

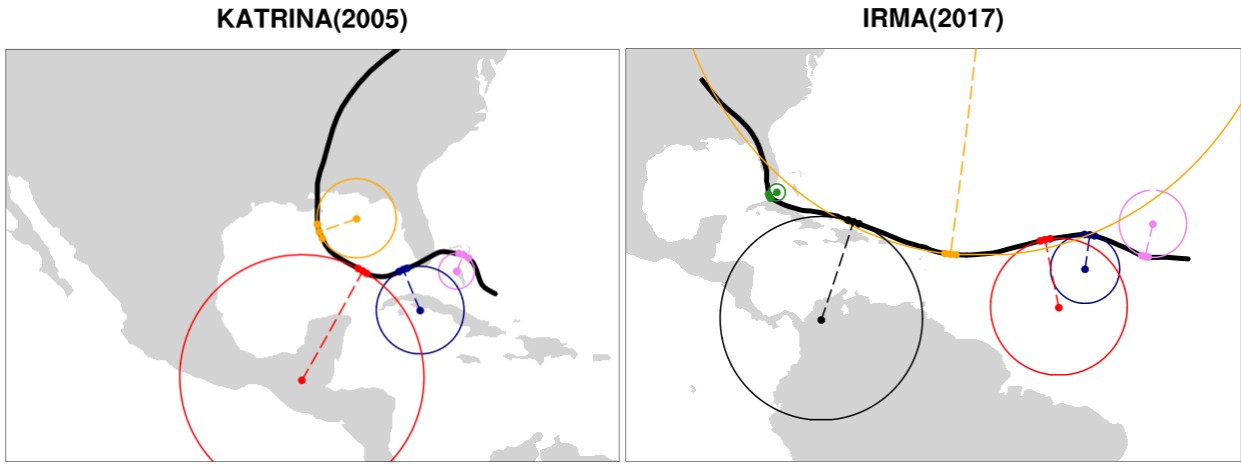

**Figure 1.** Illustration of the circle-fit and radius of curvature calculations at five selected locations along the track of hurricanes Katrina (2005) and Irma (2017).

While the expressions above were derived independently for this work, we make no claim of originality since they are based on elementary principles of geometry. Figure 1 shows some examples of the radius of curvature calculation for hurricanes Katrina (2005) and Irma (2017).

## 4 Basic Statistics

Over the years 1966–2019, 689 storms in the North Atlantic meet the 3-day threshold. Figure 2 shows some basic statistics for speed and accelerations as a function of latitude for Atlantic storms. Tropical cyclone locations are sorted in $10°$-wide latitude bins and all instances classified as non-tropical were excluded. Table 1 provides the same data. The average speed of all North Atlantic tropical cyclones (including over-land) is about 21 km/hr. As expected, tropical cyclone speed clearly varies



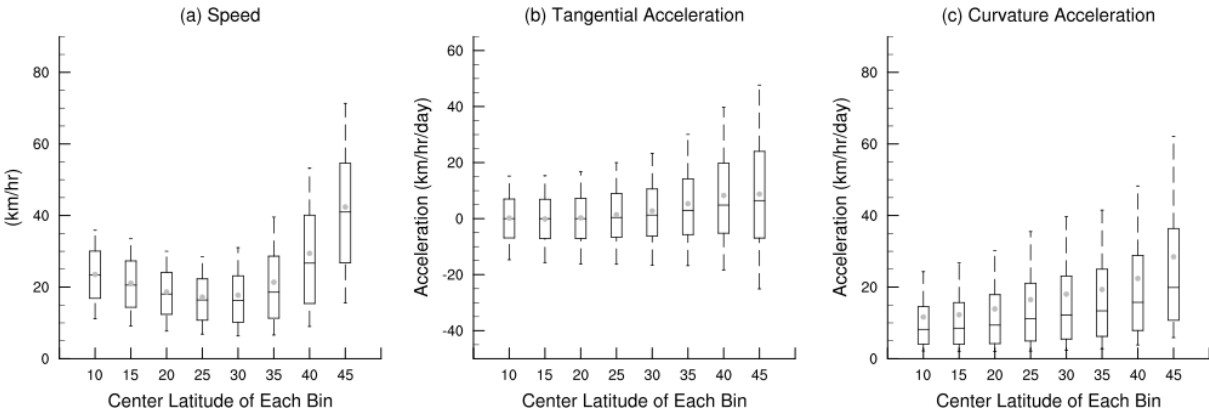

**Figure 2.** Distribution of (a) Speed(km hr$^{-1}$) ; (b) Tangential acceleration (km hr$^{-1}$ day$^{-1}$) and (c) Curvature acceleration (km hr$^{-1}$ day$^{-1}$) of Atlantic TCs as a function of latitude. Storm instances classified as ET or NR were excluded. Data from 1966–2019 was binned within $10^{o}$-wide overlapping latitude bins. Statistics shown are: median (horizontal line within the box), mean (dot), and 10th, 25th, 75th and 90th percentiles.

with latitude. It is lower in the subtropics as compared to other latitudes. It increases sharply in the vicinity of $40^{o}$ N. The

tangential acceleration, as defined in eq. 1, can be positive or negative. The mean and median tangential acceleration remains near-zero equatorward of 30°N. This suggests that tropical cyclones in this region tend to translate steadily, or are equally likely to accelerate and decelerate. The tangential acceleration is substantially positive poleward of 30°N. Tropical cyclones in these latitudes are subject to rapid acceleration. For example, the mean tangential acceleration in the $30° - 40°$N latitude band is about 5.0 km/hr day$^{-1}$. The curvature acceleration, by our definition, takes only positive values and steadily increases

with latitude. The distributions of tropical cyclone speed and tangential acceleration are relatively symmetric about the median value as compared to the curvature acceleration.

## 5    Ensemble average flow

We now examine the flow pattern associated with rapid tangential and curvature acceleration of tropical cyclones. For this, storm-relative ensemble average fields are constructed using ERA-interim reanalysis over the period 1980–2016. We use the

method outlined as follows.

- – All tropical cyclone track locations are binned into 10° wide latitude strips (e.g., 10-20° N, 20-30° N, 30-40° N). Instances where storms are classified as *non-tropical* (c.f. section 2) are excluded. For brevity, only results for the 30-40° N bin are discussed here. A total of 3515 track points are identified for this latitude bin. Note that a particular tropical cyclone could appear more than once in a latitude bin at different times.

– The tangential accelerations in each bin are separated into two categories: rapid acceleration and rapid deceleration. The rapid acceleration composite is calculated using all instances of acceleration exceeding a threshold: i.e., $a \geq a_{\tau}$, where



**Table 1.** Speed (km hr$^{-1}$), tangential and curvature acceleration (km hr$^{-1}$ day$^{-1}$) of Atlantic TCs in the IBTRaCS database as a function of latitude. Storm instances classified as ET or NR were excluded. N refers to number of 3-hourly track positions in each latitude-bin over the period 1966–2019.

| Latitude | N | Speed Mean | Speed Median | Speed Std Dev. | Tang. Accel Mean | Tang. Accel Median | Tang. Accel Std Dev. | Curv. Accel Mean | Curv. Accel Median | Curv. Accel Std Dev. |
|---|---|---|---|---|---|---|---|---|---|---|
| Full Basin | 38822 | 20.85 | 18.87 | 12.2 | 2.27 | 0.68 | 19.5 | 16.42 | 10.88 | 19.0 |
| 5–15 | 5870 | 23.53 | 23.4 | 9.4 | 0.18 | 0.0 | 14.7 | 11.65 | 8.2 | 13.1 |
| 10–20 | 13087 | 21.10 | 20.6 | 9.3 | -0.13 | -0.0 | 15.3 | 12.29 | 8.5 | 13.3 |
| 15–25 | 13656 | 18.66 | 18.1 | 8.6 | 0.26 | -0.0 | 16.0 | 13.90 | 9.4 | 15.6 |
| 20–30 | 13074 | 17.21 | 16.4 | 8.6 | 1.40 | 0.4 | 17.1 | 16.48 | 11.2 | 19.0 |
| 25–35 | 13905 | 17.74 | 16.2 | 10.2 | 2.74 | 1.3 | 19.7 | 18.04 | 12.2 | 20.6 |
| 30–40 | 10815 | 21.37 | 18.6 | 13.5 | 5.33 | 3.0 | 22.8 | 19.34 | 13.4 | 21.0 |
| 35–45 | 5272 | 29.41 | 26.7 | 18.0 | 8.27 | 4.8 | 27.4 | 22.41 | 15.8 | 22.9 |
| 40–50 | 1607 | 42.37 | 41.0 | 20.8 | 8.75 | 6.4 | 34.9 | 28.45 | 19.9 | 29.0 |
| 45–55 | 329 | 55.69 | 53.9 | 19.4 | 6.81 | 6.3 | 37.8 | 36.50 | 25.9 | 38.9 |

$\tau$ refers to a specified quantile of the acceleration distribution within the latitude bin. Similarly, the rapid tangential deceleration composite is based on all instances where $a \leq a_\tau$. We tried a variety of thresholds for $\tau$ (e.g., $0.80 - 0.95$ for rapid tangential acceleration, and $0.05 - 0.20$ for rapid tangential deceleration). Our conclusions for the composites are not sensitive to the exact choice of the threshold values as long as they are sufficiently far from the median.

– We also use the same method to create categories of curvature accelerations. For this, since we only have positive values, we interpret these two quantile-based categories as rapid acceleration and near-zero acceleration respectively.

– For each category, we compute an ensemble average composite field of the geopotential field at selected isobaric levels. The composites are calculated after shifting the grids such that the centers of all storms were coincident. The centroid position of storms was used for the composite storm center, and the corresponding time was denoted as Day-0. Lag-composites were created by shifting the dates backward and forward relative to Day-0.

– For anomaly fields, we subtract the long-term synoptic climatology from the total field. The climatology is calculated for each day of the year by averaging data for that day over the years 1980-2015. This is followed by a 7-day running mean smoother. To account for diurnal fluctuations, the daily climatology is calculated for each available synoptic hour in the ERA-interim data (00, 06, 12, and 18 UTC).

– For brevity, we only show the results for $\tau = 0.9$ for rapid acceleration and $\tau = 0.1$ for rapid deceleration. For reference, within 30–40$^o$N latitude range, $\tau = 0.9$ corresponds to $a = 32$ km/hr day$^{-1}$ and the $\tau = 0.1$ corresponds $a = -18$ km/hr day$^{-1}$. For curvature acceleration, they are, respectively, 48 and 32 km/hr day$^{-1}$. The sample size of each composite was





352 ($\approx 10\%$ of the total number of track points in this latitude range. These correspond to 196 and 168 unique storms for rapid acceleration and rapid deceleration respectively.

– Statistical significance of anomaly fields shown in this section are evaluated by comparing them against 1000 composites created by randomly drawing 352 dates for each composite from the period July–October, 1980–2015. A two-tailed significance is evaluated at the 95% confidence level with the null hypothesis being that the anomalies could have resulted from a random draw.

## 5.1 Tangential Acceleration

### 5.1.1 Day 0 (reference day) composite

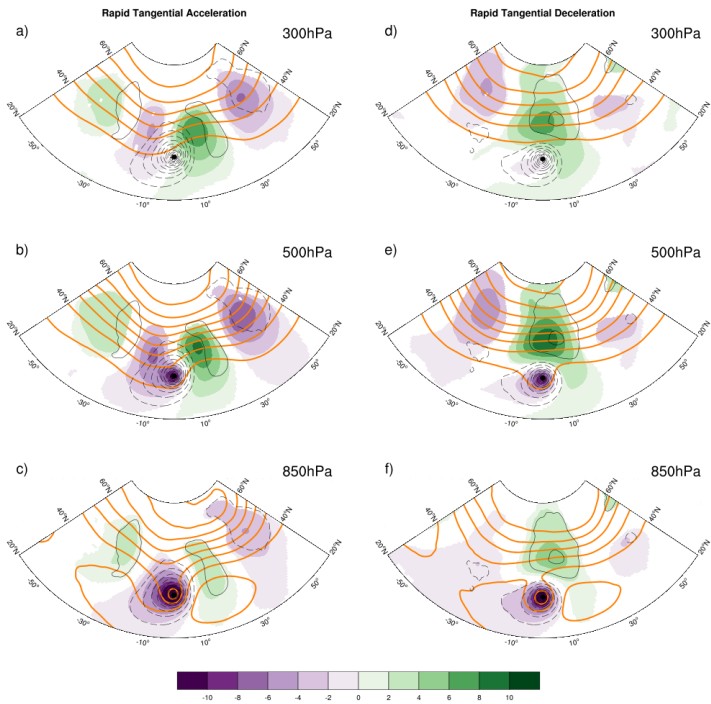

**Figure 3.** Storm-relative composite average geopotential heights (thick orange lines) and anomalies (color shaded) for all TCs located in the latitude bin 30-40$^o$N over the Atlantic. The composite fields are shown for three levels – 300 hPa, 500 hPa and 850 hPa. In each panel, the composite 1000 hPa anomalous geopotential is shown using thin black contours. All anomalies are defined relative to a long-term synoptic climatology. The contour intervals are: 12 dam, 6 dam and 3 dam for the three levels respectively. The shading interval in dam for the 300 hPa anomaly fields is shown in the label-bar. It is half of the value for the other two levels. The left column is for rapid tangential acceleration and the right column is for rapid tangential deceleration.





Figure 3 shows storm-centered composite geopotential heights (thick orange lines) and their anomalies (color shaded) for all Atlantic tropical cyclones located within 30-40° N. Two categories of tangential acceleration are shown: rapid acceleration (left column) and rapid deceleration (right column). The fields are shown at three levels – 300 hPa, 500 hPa and 850 hPa. In

each panel, the anomalous 1000 hPa geopotential is shown using thinner black contours. It highlights the composite tropical cyclone and the surface development within the extratropical stormtrack.

The ensemble average for rapid tangential acceleration (left column of Fig. 3) shows the composite tropical cyclone inter-acting with a well defined extratropical wavepacket. The tropical cyclone is straddled by an upstream trough and a downstream ridge. At 500 hPa, the geopotential anomalies of the tropical cyclone and the upstream trough are close and, consequently,

appear to be connected. This yields a negative tilt in the horizontal and indicates the onset of cyclonic wrap-up of the trough around the tropical cyclone. The 1000-hPa geopotential anomaly field is dominated by the composite tropical cyclone. It also shows the relatively weaker near-surface cyclones and anticyclones of the extratropical stormtrack. The entire wavepacket shows upshear tilt of geopotential anomalies with height, indicating baroclinic growth. This arrangement of the tropical cyclone and the extratropical wavepacket is consistent with the synoptic-scale flow that is typically associated with extratropical

transition (e.g., Bosart and Lackmann, 1995; Klein et al., 2000; McTaggart-Cowan et al., 2003; Riemer et al., 2008; Riemer and Jones, 2014; Keller et al., 2019). At this point, all storms in the ensemble were still classified as *tropical*. Thus, we interpret this composite as pre-extratropical transition completion state.

The ensemble average for rapid tangential deceleration cases (right column of Fig. 3) shows an entirely different synoptic-scale pattern. The extratropical wavepacket is substantially poleward, with a ridge immediately north of the composite tropical

cyclone. The geopotential anomalies of the extratropical wavepacket and the composite tropical cyclone appear to be distinct at all three levels, with no evidence of merger. The prominent synoptic structure is the cyclone-anticyclone dipole formed by the tropical cyclone and the extratropical ridge.

### 5.1.2 Lag Composites

To get a sense of the temporal evolution of the entire system, we show lag composites for Day-2 to Day+2 in Fig. 4. As in

the previous figure, the two categories of acceleration are arranged in the respective columns. The rows now show 500-hPa geopotential height (thick contours) and anomalies (color shaded). In each panel, the corresponding 1000-hPa geopotential height anomalies are shown by thin black contours.

The ensemble average for rapid tangential acceleration (left column of Fig. 4) shows a tropical cyclone moving rapidly towards an extratropical wavepacket. At day-2, the tropical cyclone circulation is relatively symmetric as depicted by the

contours of 1000 hPa geopotential anomalies. The downstream extratropical ridge is prominent, but the upstream trough is much weaker at this time. On Day-1, the entire extratropical wavepacket has amplified and the 500 hPa geopotential anomalies of the tropical cyclone and a developing upstream trough have merged. This process continues through Day 0. By Day+1, the composite storm has moved further poleward and eastward and is now located between the upper-level upstream trough and downstream ridge in a position that is optimal for further baroclinic development. The 1000 hPa geopotential field is now

asymmetric with a characteristic signal of a cold front.

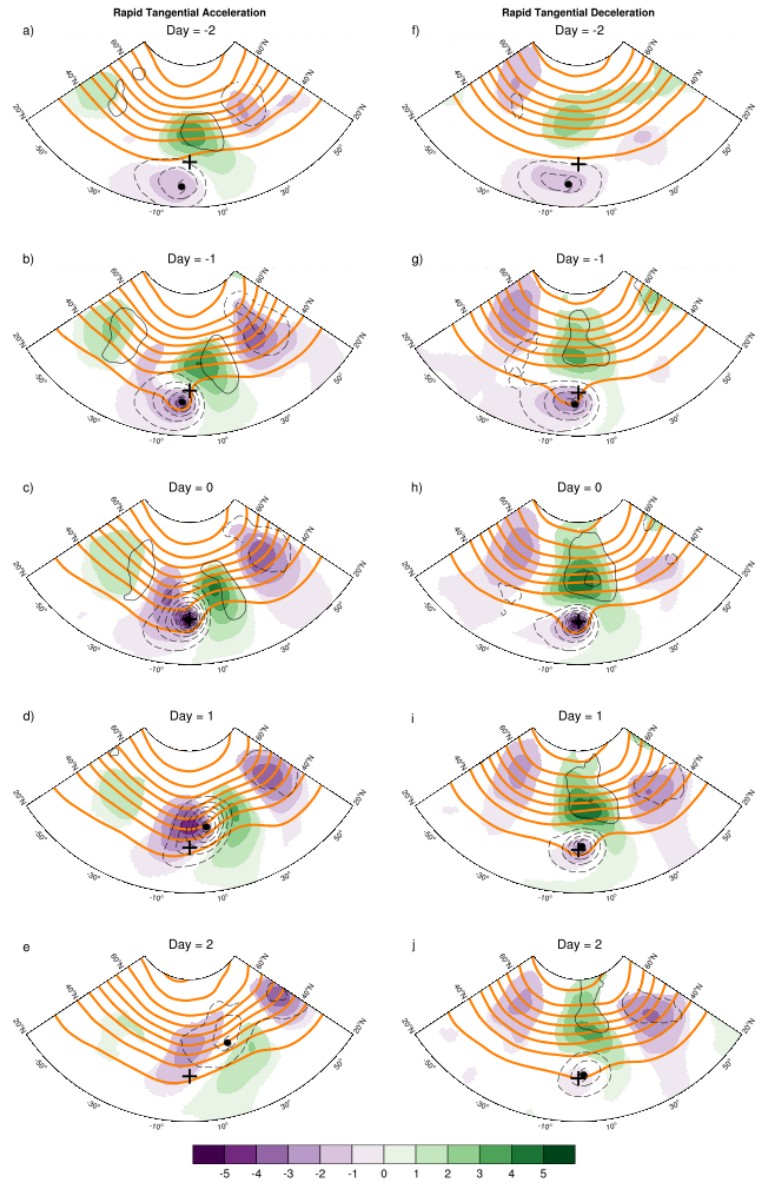

**Figure 4.** Storm-relative average 500-hPa geopotential heights (thick orange lines) and anomalies (color shaded) for all TCs located in the latitude bin 30-40$^o$N over the Atlantic. The fields are shown for lags Day-2 to Day+2. In each panel, the composite 1000 hPa anomalous geopotential is shown using thin black contours. All anomalies are defined relative to a long-term synoptic climatology. The contour interval is 6 dam and shading interval in dam is shown in the label-bar. The plus symbol shows the location of the composite TC at Day 0 and the hurricane symbol shows the approximate location at each lags. The left column is for rapid tangential acceleration and the right column is for rapid tangential deceleration



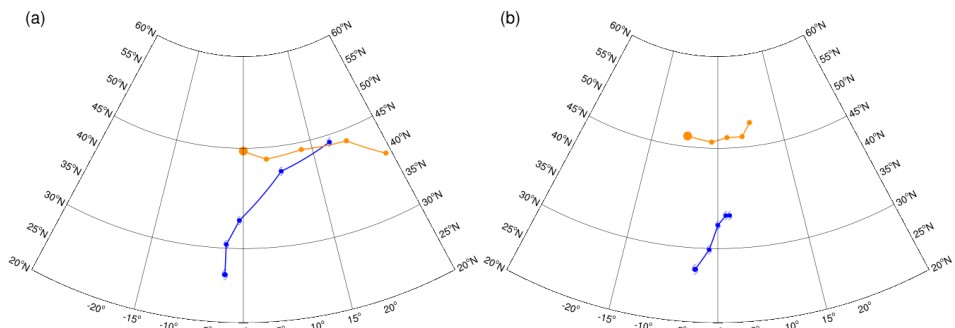

**Figure 5.** Track of the composite tropical cyclone (blue) and the downstream 500-hPa extratropical ridge (orange) from Day-2 to Day +2 for (a) rapid tangential acceleration, and (b) rapid tangential deceleration. The composites are based on all TC tracks locations within 30–40$^{o}$N. Day 0 is the reference day for the composites in Fig. 4

The picture that is evident from these 500-hPa composite fields is that, over the course of the 4 days, downstream ridge-trough couplet amplifies while simultaneously propagating eastward. The upstream trough cyclonically wraps around the tropical cyclone and the two have merged by Day 1. The geopotential gradient poleward of the storm is also enhanced, indicating a strengthening jet streak. These features are consistent with the process of extratropical transition (e.g., Keller et al., 2019). The

poleward moving tropical cyclone may either interact with an existing wavepacket or perturb the extratropical flow and excite a Rossby wavepacket that disperses energy downstream (e.g., Riemer and Jones, 2014). The outflow of the tropical cyclone is a source of low potential vorticity (PV) air that further reinforces the downstream ridge (e.g., Riemer et al., 2008).

To further illustrate the interaction, the tracks of the tropical cyclone and the 500-hPa ridge of the extratropical wavepacket are presented in Fig. 5a. It can be clearly seen that the tropical cyclone merges with the extratropical stormtrack. Furthermore,

the 500 hPa ridge has rapidly moved downstream during the 4 day period, indicating a very progressive pattern. The eastward phase speed of the extratropical wavepacket, as inferred from the track of the ridge, is $\approx 7$ ms$^{-1}$. The tropical cyclone speed, averaged over the same 4-day period, is $\approx 6$ ms$^{-1}$. The close correspondence between the two and the merger of the tracks further supports the notion that the synoptic-scale evolution during rapid acceleration cases is consistent with the canonical pattern associated with extratropical transition.

On the other hand, during rapid deceleration (right column Fig. 4), the composite tropical cyclone remains equatorward of the extratropical wavepacket and maintains a nearly symmetric structure throughout the period. The arrangement of the tropical cyclone and the extratropical ridge is akin to a vortex dipole. The extratropical wavepacket is not as progressive as in the rapid acceleration case. This is seen clearly from the tracks of the tropical cyclone and the ridge (Fig. 5b). The phase speed of the extratropical wavepacket is $\approx 3$ ms$^{-1}$ while the tropical cyclone speed is $\approx 1.5$ ms$^{-1}$. The phasing of the tropical cyclone and

the extratropical wavepacket has led to the formation of a cyclone-anticyclone vortex dipole. We return to this point and relate it to similar findings in Riboldi et al. (2019) in a later section.





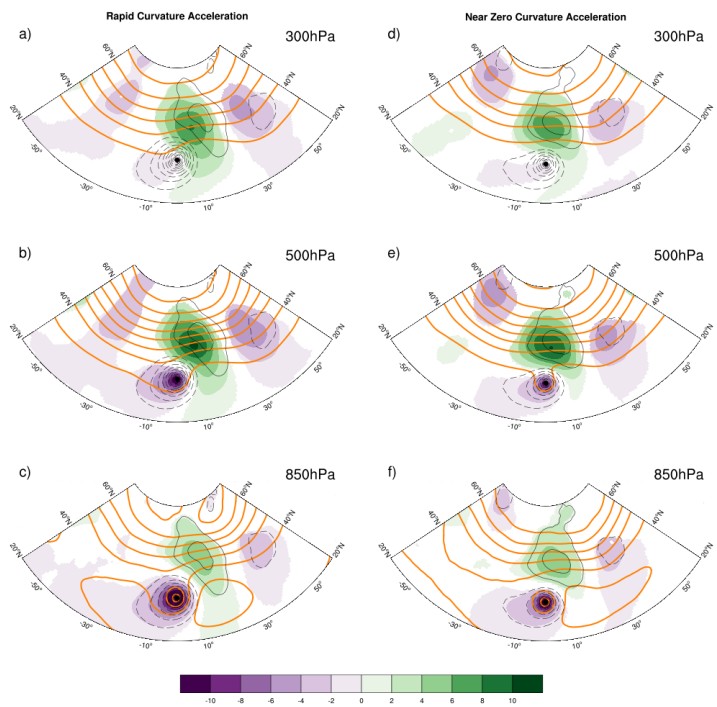

**Figure 6.** As in Fig. 3, but for rapid and near-zero curvature acceleration

## 5.2 Curvature Acceleration

### 5.2.1 Day 0 (reference day) composite

As in the previous section, Figure 6 shows storm-centered ensemble averages, but this time for the two categories of curvature acceleration. The composite for rapid acceleration (left column) shows a tropical cyclone that is primarily interacting with an extratropical ridge that is poleward and downstream of it. The upstream trough in the extratropics is weaker and farther westward as compared to the rapid tangential acceleration composite (Fig. 6a). Furthermore, instead of the upstream trough wrapping cyclonically, in this case, we see the downstream ridge wrapping anticyclonically around the tropical cyclone. This is similar to the composite 500-hPa fields that were based on recurving tropical cyclones as shown in Fig. 5 of Aiyyer (2015). Thus, in an ensemble average sense, rapid curvature acceleration appears to mark the point of the recurvature of tropical cyclones.

The composite for near-zero curvature acceleration (right column) is quite similar to the composite for rapid tangential deceleration (Fig. 6d–f). The extratropical wavepacket is poleward and the tropical cyclone-ridge system appears as a vortex dipole.



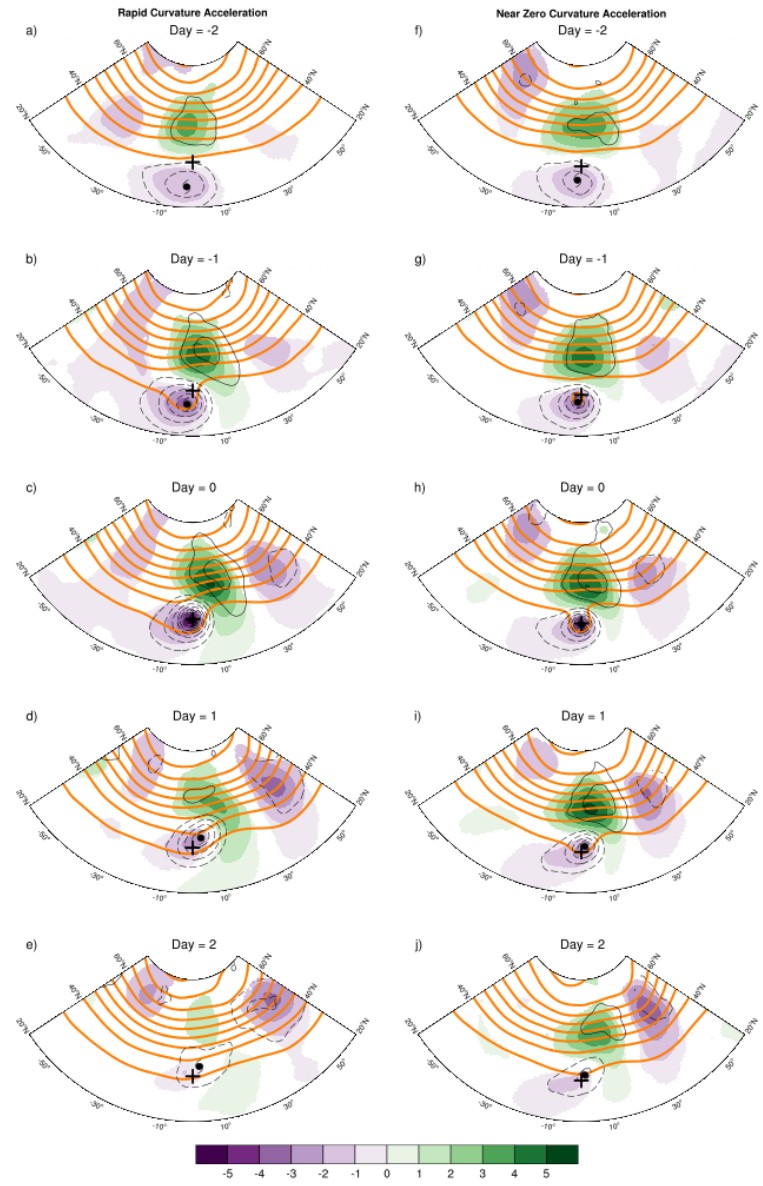

**Figure 7.** As in Fig. 4, except for rapid (left column) and near-zero (right column) curvature acceleration.

### 5.2.2 Lag Composites

The temporal evolution of the entire system for the two categories of curvature acceleration is shown in Fig. 7. For rapid curvature acceleration, we see a tropical cyclone that is moving poleward towards an extratropical ridge. During the subsequent days, the ridge moves eastward initially and begins to wrap around the tropical cyclone. This arrangement promotes the re-




curving of the tropical cyclone. By Day+2, the anticyclonic wrapping and thinning has resulted in a significantly weaker ridge

as compared to a few days prior. For the near-zero curvature cases ( Fig. 7f–g), the initial movement of the tropical cyclone
is also directly poleward towards the extratropical ridge. However, in this case, the ridge remains poleward of the tropical
cyclone. There is also significantly less anticyclonic wrapping of ridge. The tropical cyclone-ridge system takes the form of a
cyclonic-anticyclonic vortex pair similar to the rapid tangential deceleration composite.

The tracks in Fig. 8 clearly show how the tropical and extratropical systems propagate. For rapid curvature acceleration (Fig.

8a), the tropical cyclone track shows a recurving tropical cyclone. The track of the ridge confirms the initial eastward motion,
followed by a poleward shift after parts of it wrap around the tropical cyclone as noted from Fig. 7a-e. By Day+2, we do not
observe a merger of the tracks that happens in the case of rapid tangential acceleration (Fig. 5a).

The tracks for near-zero curvature acceleration (Fig. 8a) are somewhat similar to the rapid tangential deceleration (Fig. 5b).
The key point here is that, although the tropical cyclone moves poleward, tropical cyclone-ridge system acts like a vortex dipole

and is nearly stationary in the zonal direction. This arrangement of the tropical cyclone and the extratropical wavepacket is
similar to the composite fields in Fig. 10 of Riboldi et al. (2019), where they show upper-level potential vorticity (PV), 850-hPa
potential temperature and sea-level pressure. The difference is that their composite was conditioned on the acceleration of the
upstream trough for recurving western Pacific typhoons. It is, however, not surprising that we can recover a similar pattern
when we condition our composites on the basis of tropical cyclone acceleration. Since the the extratropical wavepacket and the

tropical cyclone are actively interacting, they influence each other's motion. Riboldi et al. (2019) referred to this as a phase-
locking of the upstream trough and the tropical cyclone while we have viewed this as a phase-lock between the ridge and the
tropical cyclone. The two are not mutually exclusive since the trough and the ridge are part of the same wavepacket.

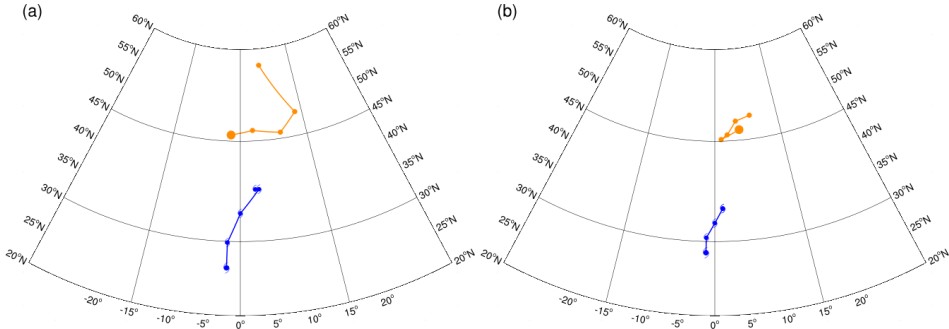

**Figure 8.** Track of the composite tropical cyclone (blue) and the downstream 500-hPa extratropical ridge (orange) from Day-2 to Day +2
for (a) rapid curvature acceleration, and (b) near-zero curvature acceleration. The composites are based on all TC tracks locations within
30–40$^o$N. Day 0 is the reference day for the composites in Fig. 7



## 6 Extratropical Transition

In the previous section, we showed that the composite synoptic-scale flow associated with rapid tangential acceleration re-
sembles a pattern that is favorable for extratropical transition. However, this does not imply that all storms in the composite
underwent extratropical transition. Some tropical cyclones may begin the process of extratropical transition but dissipate before
its completion (e.g., Kofron et al., 2010). We now consider tropical cyclone motion from a different perspective by considering
only those storms that completed the transformation from being tropical to extratropical. Hart et al. (2006) found that the time
taken for extratropical transition completion can vary considerably. For the storms that they examined, this ranged from 12–168
hours. To get a sense of the temporal evolution relative to extratropical transition completion, we examine composite tropical
cyclone speed and acceleration as a function of time. For this, we only considered those Atlantic storms during 1966–2019 that
were classified as *tropical* at some time and subsequently underwent extratropical transition. Of the 689 candidate storms that
passed the three-day threshold, 18 storms were never classified as *tropical*. Of the remaining 671 storms, 274 were eventually
classified as *extratropical*. This yields a climatological extratropical transition fraction of 41%. However, in the data record, a
few instances exist where a storm was flagged as *extratropical* earlier than *tropical*. If we remove these instances, the extra-
tropical transition fraction slightly reduces to $\approx 38\%$. These estimates are lower than the fraction of 44% during 1979–2017
in Bieli et al. (2019), and 46% during 1950-1993 in Hart and Evans (2001). The mean and median latitude of extratropical
transition completion in our data set were, respectively, 40.5°N and 41.5° N. This is consistent with Hart and Evans (2001)
who found that the highest frequency of extratropical transition in the Atlantic occurs between the latitudes of 35°N–45°N.

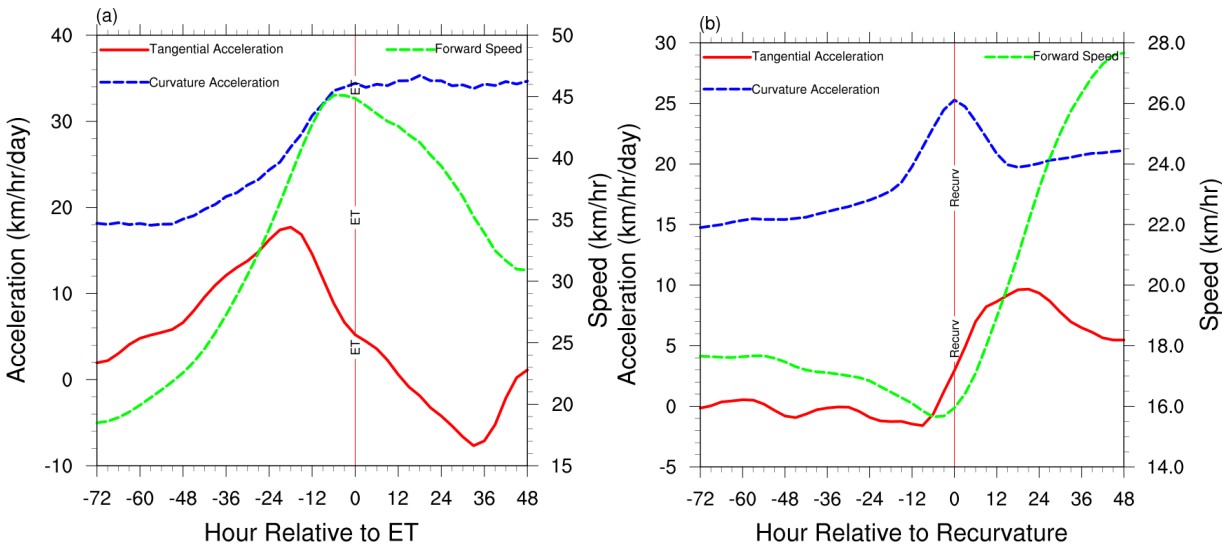

**Figure 9.** Composite speed and accelerations relative to time of (a) Extratropical transition; and (b) Recurvature. A single pass of 5-point running average was applied to the speed and tangential acceleration curves. Two passes of the same filter were applied to the curvature acceleration.





Fig. 9a shows the composite accelerations and speed relative to the time of extratropical transition. Hour 0 is defined as the first instance in the IBTrACS where the storm nature is designated as extratropical transition. We interpret this as the nearest time after extratropical transition has been completed. In an ensemble-averaged sense, the forward speed of transitioning tropical storms is seen to reach its peak around the time of extratropical transition completion. The tangential acceleration peaks about 18 hours prior to that. The curvature acceleration appears to steadily increase up to the time of extratropical transition

and stabilizes thereafter. The point here is that the peak tangential acceleration of tropical cyclones precedes extratropical transition completion. The rapid increase in the speed prior to extratropical transition completion time is a direct outcome of the interaction with the extratropical baroclinic wavepacket.

## 7    Recurvature

In the previous section, we found that the composite synoptic-scale flow associated with rapid curvature acceleration closely

matches the pattern associated with recurving of tropical cyclones (Aiyyer, 2015). To further explore this connection, Fig. 9b shows the acceleration and speed composite timeseries relative to recurvature. We follow the method described in Aiyyer (2015) to determine the location of recurvature. A total of 653 recurvature points were found for Atlantic tropical storms over the period 1966–2019. Note that a given storm could have more than one instance of recurvature. Fig 9b confirms that, in an ensemble average sense, the time of recurvature is associated with the highest curvature acceleration. Furthermore, it is also

associated with the lowest forward speed and a period of rapid increase in tangential acceleration.

## 8    Trends

We first examine the trends in the annual-mean translation speed to place our results within the context of recent studies of tropical cyclone motion. As noted in the introduction, Kossin (2018) found a decreasing trend in annual-mean tropical cyclone speed during 1949–2018 over most of the globe. That study considered all storms in the IBTrACS dataset as long as they

survived at least three days. We revisit this for the Atlantic and test the sensitivity of the trend when we exclude non-tropical systems. The rationale for this was discussed earlier in section 2. Figure 10 shows the annual-mean speed of tropical cyclones for two categories: All storms (grey) and storms excluding NR and extratropical transition designations (orange). Panel (a) shows this for the entire Atlantic and Panel (b) for the 20–40$^o$N band.

Table 2 contains the trends calculated using linear regression and the Thiel-Sen estimate. We also include the trends for

1949–2016 to compare our calculations with Kossin (2018). For 1949–2016, when we consider all tropical cyclones, the linear regression and Thiel-Sen estimates of the trends in annual-mean speed are $-0.019$ and $-0.021$ km hr$^{-1}$ year $^{-1}$. These are practically identical to the value of $-0.02$ km hr$^{-1}$ year $^{-1}$ reported by Kossin (2018). However, the trend for the satellite era over the entire basin switches to a positive value of $\approx 0.028$ km hr$^{-1}$ year $^{-1}$. The sensitivity to the choice of the years is consistent with Lanzante (2019) who showed that the negative trend of the annual-mean speed over the longer period

1949–2016 was reduced in magnitude by accounting for the change points such as those associated with the advent of satellite-

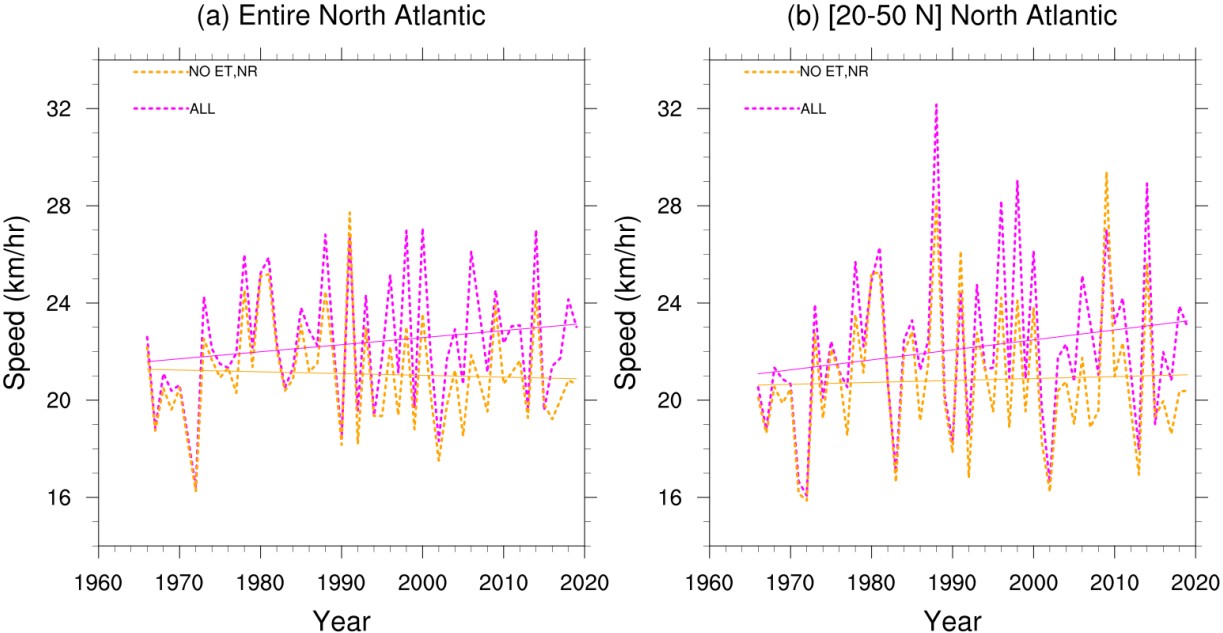

**Figure 10.** Annual-mean speed and linear trend for (a) The entire Atlantic; and (b) 20–50°N latitude band. The grey curve is for all storms in the IBTRaCS dataset while the orange curve excludes instances when the storm was classified as ET or NR.

**Table 2.** Trends in Speed (km hr$^{-1}$ year$^{-1}$). *All storms* refers to all instances of a system recorded in the IBTraCs. ET refers to storm nature designated as extratropical, while NR refers to instances when the storm nature was not recorded.

| | 1966–2019 | | | | 1949–2019 | | | | 1949–2016 | | | |
|---|---|---|---|---|---|---|---|---|---|---|---|---|
| | LR | | MK-TS | | LR | | MK-TS | | LR | | MK-TS | |
| | Trend | p-value | Trend | p-value | Trend | p-value | Trend | p-value | Trend | p-value | Trend | p-value |
| **Atlantic (All storms)** | | | | | | | | | | | | |
| Full basin | 0.029 | 0.19 | 0.028 | 0.15 | -0.016 | 0.28 | -0.016 | 0.32 | -0.019 | 0.24 | -0.021 | 0.25 |
| 20–50 | 0.041 | 0.15 | 0.035 | 0.11 | -0.011 | 0.56 | -0.019 | 0.29 | -0.012 | 0.55 | -0.023 | 0.26 |
| **Atlantic (Excluding ET,NR)** | | | | | | | | | | | | |
| Full basin | -0.007 | 0.70 | -0.008 | 0.62 | -0.004 | 0.77 | -0.007 | 0.48 | -0.002 | 0.90 | -0.006 | 0.63 |
| 20–50°N | 0.008 | 0.76 | 0.002 | 0.93 | <0.001 | 1.00 | -0.009 | 0.52 | 0.005 | 0.79 | -0.007 | 0.72 |

based weather monitoring. When we remove non-tropical data, the trends for various periods and regions are generally lower. Furthermore, none of the trends shown in Table 2 can be deemed significant if we use a p-value of 0.05 as the cut-off. We return to this point in the following section.

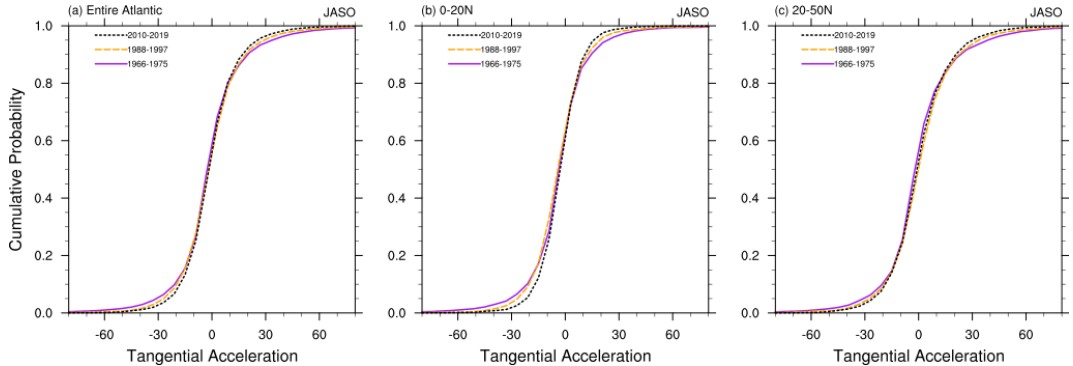

**Figure 11.** Cumulative distributions of tangential acceleration (km hr$^{-1}$ day$^{-1}$) for July-October, 1966–2019 for (a) The entire Atlantic; (b) 0–20$^o$N; and (c) 20–50$^o$N

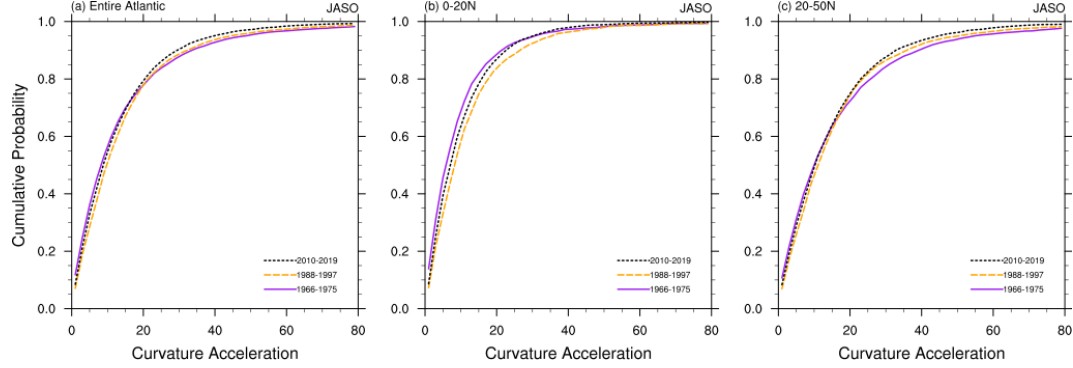

**Figure 12.** Cumulative distributions of curvature acceleration (km hr$^{-1}$ day$^{-1}$) for July-October, 1966–2019 for (a) The entire Atlantic; (b) 0–20$^o$N; and (c) 20–50$^o$N

## 8.1 Quantile regression

Figure 11 compares the cumulative probability distribution (CDF) of tangential accelerations over three 10-year periods: 1966–1975, 1988-1997, and 2010–2019. The data covers the peak hurricane season: July–October (JASO). Three Atlantic regions are shown - the entire basin, 0–20$^o$N, and 20–50$^o$N. In all three CDFs, the lower and upper-tail probability of the distribution appear to show a shift towards the median. The direction of the shift indicates a reduction in the frequency of both rapid tangential acceleration ($a \geq 15$ km/hr day$^{-1}$) and rapid tangential deceleration ($a \leq -15$ km/hr day$^{-1}$) from the earlier to

recent decades. This is most pronounced over the 20–50$^o$N latitude band. The CDFs for curvature acceleration (bottom row of Fig. 12) show a similar shift towards less frequent rapid acceleration. The CDF over the entire year shows similar shifts. When we consider a smaller subset of months, we find that the shifts are more pronounced when we omit October and November (not shown).





In the preceding sections, we showed that rapid acceleration or deceleration of tropical cyclones are typically associated
with interactions with the extratropical baroclinic stormtrack. The attendant synoptic-scale pattern are distinct in the phasing
of the tropical cyclone and the extratropical wavepacket. It is of interest to determine if the shift in CDFs of acceleration (Figs.
11, 12) are related to long-term trends. The motivation being that it can inform us about potential changes in the nature of
tropical cyclone-baroclinic stormtrack interaction. Given that we are interested in the long-term trends of rapid acceleration
and deceleration – i.e., the tails of the probability distribution – we use quantile regressions (QR) as developed by Koenker
and Bassett (1978). QR is a useful tool to model the behavior of the entire probability distribution and has been used in diverse
fields and applications (e.g., Koenker and Hallock, 2001). In atmospheric sciences, QR has been applied to examine trends
in extreme precipitation and temperature (e.g., Koenker and Schorfheide, 1994; Barbosa et al., 2011; Gao and Franzke, 2017;
Lausier and Jain, 2018; Passow and Donner, 2019).

The standard form of the simple linear regression model for a response variable $Y$ in terms of its predictor $X$ is written as:

$$\mu\{Y|X\} = \beta_o + \beta_1 X \tag{5}$$

Where, $\mu\{Y|X\}$ is the conditional mean of Y given a variable X, and $\beta_o$ and $\beta_1$ are, respectively, the intercept and the slope.
This linear model fits the mean of a response variable under the assumption of constant variance over the entire range of the
predictor. However, when the data is heteroscedastic, and there is interest in characterizing the entire distribution - and not just
the mean - QR is more appropriate and insightful. The standard form of QR is written as follows (e.g., Lausier and Jain, 2018):

$$Y(\tau|X) = \beta_o^{(\tau)} + \beta_1^{(\tau)} X + \epsilon^{(\tau)} \tag{6}$$

where $Y(\tau|X)$ denotes the conditional estimate of Y at the quantile $\tau$ for a given X. By definition, $0 < \tau < 1$. In our case,
Y is the timeseries vector of either acceleration or speed, and X is the vector comprised of the dates of the individual storm
positions. Here, $\beta_o^{(\tau)}$ and $\beta_1^{(\tau)}$ denote the intercept and slope, while the $\epsilon^{(\tau)}$ denotes the error. As noted in previous studies
cited above, QR does not make any assumption about the distribution of parameters and is known to be relatively robust to
outliers. To determine the trends, we fit the quantile regression model for a range of quantiles between .05 and .95. Instead
of calculating annual averages to get one value of acceleration or speed per year, we retain all of the individual values for the
tropical cyclones. The corresponding time for each data point is assigned as a fractional year.

Figure 13 shows the results of QR for tangential acceleration. The panels on the left show the acceleration (light blue
circles) at individual track locations from 1966-2019 (ET and NR excluded). The dashed magenta lines are the linear fits for
the quantiles ranging from 0.5 to 0.95. The panels on the right show the slope (trend; km/hr day$^{-1}$ year$^{-1}$) of the linear line as
a function of the quantile. These figures include the best fit using ordinary least squares (OLS; red line) that models the mean
of the distribution. Also included are the associated 95% confidence bounds. The top row includes data from all months over
the entire Atlantic. The middle row is for 20–50$^o$N over the peak tropical cyclone months (July-October), and the bottom row
restricts the data to August-September. The latter two illustrate some of the sensitivity to the choice of domain and months
of analysis. They also focus our attention on the region where tropical cyclones are most likely to interact with extratropical



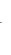


**Figure 13.** Quantile regressions of tangential acceleration for regions and months shown on the panels: The **left** columns show the acceleration (light blue circles) for all TCs (NR and ET excluded) as a function of time. The dotted magenta lines show the linear fits quantiles ranging from 0.05 to 0.95. The red line shows the ordinary least square fit for the mean. The **right** columns show the estimates of the slope (i.e., the trend in km/hr day$^{-1}$ year$^{-1}$) for each quantile, along with the 95% confidence band (dotted magenta). Also shown is the ordinary least square estimate of trend (red line) and its 95% confidence band for each quantile.

systems. The corresponding numerical values are shown in Table 3. Recall from Table 1 that the median value ($\tau = .5$) of





**Table 3.** Quantile trends of tangential acceleration over 1966-2019 (km hr$^{-1}$ day$^{-1}$ year$^{-1}$) for months and regions labeled below. Trends of magnitude below 0.01 are not reported.

| | Entire Atlantic All Months | | | | 20–50$^o$N Jul-Oct | | | | 20–50$^o$N Aug-Sep | | | |
|---|---|---|---|---|---|---|---|---|---|---|---|---|
| $\tau$ | Trend | Change | p | 95% Conf | Trend | Change | p | 95% conf. | Trend | Change | p | 95% conf. |
| OLS | 0.01 | – | 0.01 | 0.00, 0.03 | 0.01 | 10 | 0.50 | -0.01, 0.02 | , – | 4 | 0.83 | -0.01, 0.02 |
| 0.05 | 0.12 | 23 | <0.01 | 0.09, 0.16 | 0.07 | 13 | <0.01 | 0.02, 0.11 | 0.14 | 25 | <0.01 | 0.09, 0.20 |
| 0.10 | 0.06 | 18 | <0.01 | 0.04, 0.08 | 0.03 | 10 | 0.03 | 0.00, 0.07 | 0.07 | 19 | <0.01 | 0.03, 0.10 |
| 0.15 | 0.03 | 12 | <0.01 | 0.01, 0.05 | – | 0 | 0.98 | -0.02, 0.02 | 0.01 | 5 | 0.35 | -0.02, 0.04 |
| 0.20 | 0.02 | 9 | 0.02 | <0.01, 0.03 | – | -2 | 0.76 | -0.02, 0.02 | – | 2 | 0.70 | -0.02, 0.03 |
| 0.30 | 0.01 | 8 | 0.19 | -0.00, 0.02 | — | 4 | 0.60 | -0.01, 0.02 | 0.01 | 6 | 0.53 | -0.01, 0.02 |
| 0.50 | 0.03 | – | <0.01 | 0.02, 0.03 | 0.03 | – | <0.01 | 0.02, 0.04 | 0.01 | – | 0.04 | 0.00, 0.03 |
| 0.70 | 0.02 | 18 | <0.01 | 0.01, 0.03 | 0.03 | 19 | <0.01 | 0.01, 0.05 | – | 2 | 0.75 | -0.02, 0.02 |
| 0.80 | 0.02 | 9 | 0.01 | 0.00, 0.04 | 0.02 | 9 | 0.04 | 0.00, 0.05 | -0.01 | -5 | 0.40 | -0.04, 0.02 |
| 0.85 | – | -2 | 0.73 | -0.02, 0.02 | 0.01 | 1 | 0.64 | -0.02, 0.04 | -0.04 | -11 | 0.05 | -0.07, -0.00 |
| 0.90 | -0.02 | -5 | 0.13 | -0.04, 0.01 | -0.03 | -7 | 0.09 | -0.07, 0.00 | -0.07 | -15 | <0.01 | -0.12, -0.02 |
| 0.95 | -0.11 | -18 | <0.01 | -0.15, -0.07 | -0.14 | -18 | <0.01 | -0.20, -0.07 | -0.17 | -23 | <0.01 | -0.25, -0.10 |

tangential acceleration is a small positive number. As such, $\tau < 0.5$ is indicative of deceleration, while $\tau \geq 0.5$ is indicative of acceleration.

From Fig. 13 and Table 3, we note that the OLS estimate of the trend is weakly positive when data from all months over the entire Atlantic are considered. However, the OLS estimate is not statistically significant for the 20–50$^o$N region. As expected, the regression quantiles show a fuller picture. The slopes of the individual quantiles provide an estimate of the trends of the specific portions of the probability distribution. The key finding here is that the magnitudes of both rapid deceleration and rapid acceleration show a statistically-significant reducing trend. This is reflected in the positive slope for $\tau \leq 0.15$ and negative slope

for $\tau \geq 0.85$ (Table 3 and left columns of Fig. 13 ). It also appears that the trends for $\tau < 0.5$ are generally positive, implying a reduction in the magnitude of tangential deceleration at all quantile thresholds. On the other hand, the positive slopes seen for $0.5 < \tau < 0.8$ suggest that there is an increasing trend in the values of tangential acceleration that are closer to the median. This shift towards less-extreme acceleration is noted in all three regional categories, albeit with varying degree of statistical significance. The trends in these quantiles are, however, weaker compared with those of the tails. From the right column of

Fig. 13, it is clear that the trends of the tails of the distribution are significant and fall outside the 95% confidence bounds of the OLS estimate of the trend.

    The QR results for curvature acceleration (Fig. 14 and Table 4) show statistically significant, weak positive trends for $0 < \tau < 0.5$. However, the trends switch to increasingly negative values above the median. As in the case of tangential acceleration,



**Figure 14.** As in Fig. 13, but for curvature acceleration.

the decelerating trends in the upper quantiles of the distribution ($\tau \geq 0.8$) are statistically significant and outside the 95%
confidence bounds of the OLS estimate of the trend in the mean.

For completeness, we also show the corresponding QR results for translation speed (Fig. 15 and Table 5). The OLS estimate
of the trend is nearly the same value as it was for the annual-mean speeds. However, it is now statistically significant. The
change in the p-value reflects the fact that the sample size is much higher since the data is not averaged annually. When we





**Table 4.** Quantile trends of curvature acceleration over 1966-2019 (km hr$^{-1}$ day$^{-1}$ year$^{-1}$) for months and regions labeled below. Trends of magnitude below 0.01 are not reported.

| | Entire Atlantic | | | | 20–50$^o$N | | | | 20–50$^o$N | | | |
| | All Months | | | | Jul-Oct | | | | Aug-Sep | | | |
| $\tau$ | Trend | Change | p | 95% Conf | Trend | Change | p | 95% conf. | Trend | Change | p | 95% conf. |
|---|---|---|---|---|---|---|---|---|---|---|---|---|
| OLS | -0.02 | -5 | <0.01 | [-0.03, -0.01] | -0.04 | -11 | <0.01 | [-0.06, -0.02] | -0.06 | -15 | <0.01 | [-0.08, -0.04] |
| 0.05 | 0.01 | 42 | <0.01 | 0.01, 0.01 | 0.01 | 40 | <0.01 | 0.00, 0.01 | 0.01 | 27 | 0.04 | 0.00, 0.01 |
| 0.10 | 0.01 | 32 | <0.01 | 0.01, 0.02 | 0.01 | 25 | <0.01 | 0.00, 0.02 | 0.01 | 16 | 0.05 | 0.00, 0.01 |
| 0.15 | 0.01 | 23 | <0.01 | 0.01, 0.02 | 0.01 | 14 | 0.01 | 0.00, 0.01 | <0.01 | 7 | 0.25 | -0.00, 0.01 |
| 0.20 | 0.01 | 17 | <0.01 | 0.01, 0.02 | 0.01 | 13 | <0.01 | 0.00, 0.02 | 0.01 | 7 | 0.22 | -0.00, 0.01 |
| 0.30 | 0.02 | 15 | <0.01 | 0.01, 0.02 | 0.01 | 10 | 0.01 | 0.00, 0.02 | 0.01 | 7 | 0.11 | -0.00, 0.02 |
| 0.50 | 0.02 | 8 | <0.01 | 0.01, 0.03 | <0.01 | 1 | 0.55 | -0.01, 0.02 | -0.01 | -4 | 0.37 | -0.02, 0.01 |
| 0.70 | – | 0 | 0.72 | -0.01, 0.02 | -0.03 | -7 | 0.01 | -0.05, -0.01 | -0.04 | -11 | <0.01 | -0.06, -0.02 |
| 0.80 | -0.03 | -7 | <0.01 | -0.05, -0.01 | -0.06 | -11 | <0.01 | -0.08, -0.03 | -0.08 | -15 | <0.01 | -0.11, -0.05 |
| 0.85 | -0.06 | -11 | <0.01 | -0.08, -0.03 | -0.09 | -14 | <0.01 | -0.13, -0.05 | -0.12 | -20 | <0.01 | -0.16, -0.08 |
| 0.90 | -0.09 | -13 | <0.01 | -0.12, -0.06 | -0.12 | -15 | <0.01 | -0.16, -0.07 | -0.18 | -22 | <0.01 | -0.23, -0.12 |
| 0.95 | -0.15 | -15 | <0.01 | -0.20, -0.09 | -0.22 | -20 | <0.01 | -0.31, -0.13 | -0.33 | -29 | <0.01 | -0.43, -0.22 |

**Table 5.** Quantile trends of translation speed over 1966-2019 (km hr$^{-1}$ year$^{-1}$) for months and regions labeled below. Trends of magnitude below 0.01 are not reported.

| | Entire Atlantic | | | | 20–50$^o$N | | | | 20–50$^o$N | | | |
| | All Months | | | | Jul-Oct | | | | Aug-Sep | | | |
| $\tau$ | Trend | Change | p | 95% Conf | Trend | Change | p | 95% conf. | Trend | Change | p | 95% conf. |
|---|---|---|---|---|---|---|---|---|---|---|---|---|
| OLS | -0.01 | -2 | 0.05 | -0.01, 0.0 | -0.01 | -3 | 0.04 | -0.02, -0.0 | -0.04 | -9 | <0.01 | -0.05, -0.02 |
| 0.05 | – | -13 | <0.01 | -0.02, -0.01 | -0.02 | -17 | <0.01 | -0.03, -0.01 | -0.02 | -17 | <0.01 | -0.03, -0.01 |
| 0.10 | – | -6 | 0.02 | -0.01, -0.00 | -0.01 | -8 | 0.02 | -0.02, -0.00 | -0.01 | -9 | 0.04 | -0.02, -0.00 |
| 0.15 | – | -2 | 0.46 | -0.01, 0.00 | – | -1 | 0.98 | -0.01, 0.01 | – | -1 | 0.93 | -0.01, 0.01 |
| 0.20 | – | -2 | 0.52 | -0.01, 0.01 | – | -2 | 0.66 | -0.01, 0.01 | – | -2 | 0.73 | -0.01, 0.01 |
| 0.30 | – | -1 | 0.61 | -0.01, 0.01 | -0.01 | -3 | 0.25 | -0.02, 0.00 | – | -7 | 0.01 | -0.03, -0.00 |
| 0.50 | – | -1 | 0.99 | -0.01, 0.01 | -0.01 | -5 | 0.01 | -0.02, -0.00 | -0.04 | -12 | <0.01 | -0.05, -0.03 |
| 0.70 | – | -2 | 0.34 | -0.01, 0.01 | -0.01 | -3 | 0.18 | -0.02, 0.00 | -0.04 | -10 | <0.01 | -0.06, -0.03 |
| 0.80 | -0.01 | -2 | 0.32 | -0.02, 0.01 | -0.02 | -5 | 0.02 | -0.04, -0.00 | -0.07 | -13 | <0.01 | -0.09, -0.05 |
| 0.85 | -0.03 | -5 | <0.01 | -0.04, -0.01 | -0.03 | -5 | 0.01 | -0.05, -0.01 | -0.09 | -15 | <0.01 | -0.12, -0.07 |
| 0.90 | -0.03 | -4 | <0.01 | -0.04, -0.01 | – | – | 0.96 | -0.03, 0.03 | -0.09 | -14 | <0.01 | -0.13, -0.06 |
| 0.95 | 0.02 | 2 | 0.20 | -0.01, 0.05 | -0.01 | -2 | 0.70 | -0.07, 0.04 | -0.08 | -10 | <0.01 | -0.14, -0.03 |





**Figure 15.** As in Fig. 13, but for translation speed.

consider the entire Atlantic, estimated trends from QR are nearly the same as the OLS trend with the exception of $\tau = 0.95$.
However, when we consider the subsets of the data for $20$–$50^{o}N$, there are some notable differences. In particular, for August-
September, the trends in the fastest translation speeds are even more negative.

Tables 3, 4 and 5 also include the percent changes defined using the first and last value in the linear fit over the period
1966–2019. If we subjectively assume that a statistically significant change of magnitude at least 10% over the past 54 years





can be deemed *robust*, then the key outcome of the QR is the following: The trends in the tails of the distribution of the
accelerations and speeds are most robust for the August-September months. For the extratropical region ($20$–$50^{o}N$), both
rapid tangential acceleration and deceleration show robust reductions. This indicates a general narrowing of the tangential
acceleration distribution over time. The curvature acceleration shows an increase for the lower quantiles and reduction for
the upper. This suggests a shift in the curvature acceleration towards smaller values, consistent with the results for tangential
acceleration. Forward speed shows mostly reducing trends for both upper and lower tails, indicating that extremes in speeds
are reducing over time.

## 9 Discussion

Ensemble-average composites of atmospheric fields show distinct synoptic-scale patterns when they are categorized on the
basis of the acceleration of tropical cyclones. The composites for rapid tangential acceleration outside the deep tropics depict
a synoptic-scale pattern that is consistent with the extratropical transition of tropical cyclones. This is unsurprising since it is
generally known that tropical storms speed up during extratropical transition. The novel aspect here is that we have recovered
the signal of extratropical transition from the perspective of acceleration. The composites show a poleward moving tropical
cyclone that is straddled by an upstream trough and a downstream ridge. Subsequently, the tropical cyclone merges with the
extratropical wavepacket ahead of the trough in an arrangement that is conducive for further baroclinic development. Features
commonly associated with extratropical transition such as the downstream ridge-building, amplification of the upper-level jet
streak and downstream development can be clearly seen in the composite maps (Fig. 3 and Fig. 4a-e). The composites for
rapid curvature acceleration also show the impact of the phasing of the tropical cyclone and the extratropical wavepacket. For
this category, we recover a synoptic-scale pattern that is similar to the one obtained in composites based on the recurvature of
tropical cyclones (Aiyyer, 2015).

In contrast, the composite fields for rapid tangential and curvature deceleration show a tropical cyclone that approaches an
extratropical ridge. The upstream trough remains at a distance and the tropical cyclone does not merge with the extratropical
wavepacket, but instead remains equatorward of it over the following few days. The tropical cyclone and the extratropical
ridge – at least locally – can be viewed as a vortex dipole. The combined system remains relatively stationary compared to
the progressive pattern for rapid acceleration. This arrangement qualitatively resembles a dipole block — an important mode
of persistent anomalies in the atmosphere (e.g., McWilliams, 1980; Pelly and Hoskins, 2003). The canonical dipole block is
depicted as a vortex pair comprised of a warm anticyclone and a low-latitude cut-off cyclone (e.g., Haines and Marshall, 1987;
McTaggart-Cowan et al., 2006). The dynamics of blocked flows are rich and the subject of a variety of theories that are far
from settled (e.g. Woollings and Barriopedro, 2018). In the present case, the slowly propagating cyclone-anticyclone pair is
likely an outcome of a fortuitous phasing of the tropical cyclone and the extratropical ridge.

Our study takes a complementary view of the extratropical interaction described in Riboldi et al. (2019). In particular, we
note the close correspondence between their composite sea-level pressure and potential vorticity composites (their Fig. 10) for
decelerating troughs and our geopotential composites for rapidly decelerating tropical cyclones (right columns of Fig. 3 and



Fig. 6). The vortex dipole can be seen in all three figures. Riboldi et al. (2019) conditioned their composites on the basis of the acceleration of the upstream trough interacting with recurving typhoons in the western North Pacific. We recover the same pattern when we condition the composites based on rapidly decelerating tropical cyclones in the North Atlantic.

The interaction between the tropical cyclone and the extratropical wavepacket that leads to the deceleration of the entire system can be viewed from a PV perspective. As noted by Riboldi et al. (2019), both adiabatic and diabatic pathways are active in this interaction. In the former, the induced flow from the cyclonic vortex (tropical cyclone) will be westward within the poleward anticyclonic vortex (ridge). The induced flow from the ridge will also be westward at the location of the tropical cyclone. The combined effect will be a mutual westward advection, and thus reduced eastward motion in the earth-relative

frame. The latter, diabatic pathway relies on the amplification of the ridge through the action of precipitating convection in the vicinity of the tropical cyclone. The negative vertical gradient of diabatic heating in upper levels of the tropical cyclone implies that its anticyclonic outflow is a source of low PV air (e.g., Wu and Emanuel, 1993). The advection of this low PV air by the irrotational component of the outflow and its role in ridge-building has been extensively documented (e.g., Atallah et al., 2007; Riemer et al., 2008; Grams et al., 2013; Archambault et al., 2015). Riboldi et al. (2019) also showed that the ridge is more

amplified for rapidly decelerating troughs as compared to accelerating troughs. They implicated stronger latent heating and irrotational outflow for this difference. Our composites also show a stronger ridge for rapid tropical cyclone deceleration as compared to rapid acceleration. This can be noted by comparing the left and right columns of Figs. 4 and 7. The amplification of the ridge also results in the slow-down of the upstream trough, and as shown by Riboldi et al. (2019) yields frequent downstream atmospheric blocking events.

The tracks of TC in the vicinity of extratropical wavetrains and subtropical ridges are sensitive to the existence of bifurcation points (e.g., Grams et al., 2013; Riemer and Jones, 2014), and small shifts in positions can yield different outcomes for motion and extratropical transition. Bifurcation points also exist in the case of tropical-cyclone cut-off low interactions (Pantillon et al., 2016). Our acceleration-based composites have further highlighted the impact of the phasing of the tropical cyclone and the extratropical wavepacket in mediating the interactions between them.

While there is some storm-to-storm variability, in an average sense the tangential acceleration peaks 18 hours prior to completion of extratropical transition. Interestingly, the forward speed peaks around the time of completion of extratropical transition. Curvature acceleration increases rapidly prior to extratropical transition and remains nearly steady after this time. Composite time series also show that the curvature acceleration peaks at track recurvature while the forward speed is nearly at its minimum. The tangential acceleration shows a sharp, steady increase around recurvature. This is consistent with the

observations that extratropical transition is typically completed within 2-4 days of recurvature. A related relevant question is the following: Is rapid tangential acceleration a sign of imminent extratropical transition? We found that $\approx 65\%$ of the storms that comprised the composites for rapid acceleration (Left column: Fig. 3) completed the transition within 3 days of the reference time (day 0). This is substantially higher than the climatological fraction of $\approx 50\%$ for extratropical transition over the entire basin and storm lifetime. On the other hand, only $\approx 39\%$ of the storms that comprised the composites for rapid

deceleration (Right column: Fig. 3) completed the transition within a similar time range. This fraction is substantially lower than the climatological fraction. It is, however, consistent with the observation made earlier that the synoptic-scale pattern





for rapid deceleration promotes recurvature rather than extratropical transition. Furthermore, not all recurving storms become extratropical (Evans et al., 2017).

The composites show that rapid acceleration and deceleration can be viewed as a proxy for distinct types of tropical cyclone-
extratropical interactions. It is of interest, therefore, to ascertain whether any trends in tropical cyclone accelerations can be found in the track data. For this, we use quantile regression to examine the linear trends over the entire distribution. The tails of acceleration and translation speed show statistically significant trends. The trends for the extratropical regions of the Atlantic (20–50$^o$N) are most robust for August-September. Both rapid tangential acceleration and deceleration have reduced over the past five decades. The trends for curvature acceleration show increases for the lower quantiles and decreases for
the upper quantiles. The forward speed, particularly values above the median, also shows robust decreases for the August-September months. This supports the general conclusion of Kossin (2018) that tropical cyclones have slowed down in the past few decades.

We have not explored the physical basis for the trends discussed above. We, however, speculate that they are indicative of systematic changes in the interaction between tropical cyclones and extratropical waves. It is, however, unclear from our
preliminary examination whether the trends reflect changes in the frequency or some measure of the *strength* of the interactions. We also recognize that the notion of strength of interaction needs a firm and objective definition. Nevertheless, there are some recent findings related to the atmospheric general circulation that may be relevant to this point. First, there is growing evidence for a poleward shift in the stormtrack of extratropical baroclinic eddies. As noted by Tamarin and Kaspi (2017) and references therein, this shift has been found in both reanalysis data and climate model simulations. Separately, Coumou et al. (2015) found
a robust decrease of the zonal wind and the amplitude of synoptic-scale Rossby waves in the ERA-interim reanalysis over the Northern hemisphere during the months June-August. We hypothesize that the poleward shift of the extratropical waves and their weakening could potentially account for the acceleration trends reported here. This, however, needs to be examined further if any robust conclusion regarding attribution to climate change is to be made.

## 10 Conclusions

When we separate tropical cyclones on the basis of their acceleration and consider ensemble average composites of atmospheric fields (e.g., geopotential), we get two broad sets of synoptic-scale patterns. The composite for rapid tangential acceleration shows a poleward moving tropical cyclone straddled by an upstream trough and a downstream ridge. The subsequent merger of the tropical cyclone and the developing extratropical wavepacket is consistent with the process of extratropical transition. The composite for rapid curvature acceleration shows a prominent downstream ridge that promotes recurvature. On the other
hand, the synoptic-scale pattern for rapid tangential as well as curvature deceleration takes the form of a cyclone-anticyclone dipole with a ridge directly poleward of the tropical cyclone. We note the qualitative resemblance of this arrangement with the canonical dipole block. Some of our findings from the perspective of tropical cyclone acceleration closely match those of Riboldi et al. (2019), who conditioned their diagnostics on the basis of the acceleration of the upstream trough and recurving western north Pacific typhoons.



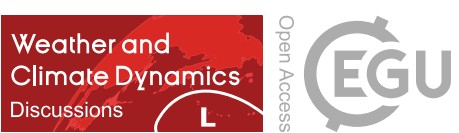

Accelerations and speed show robust trends in the tails of their distribution. For the extratropical region of the Atlantic (20–50$^o$N), and particularly for the months August-September, peak acceleration/deceleration, as well as speeds of tropical cyclones, have reduced over the past 5 decades. The reduction in the tails of the speed distribution provide complementary evidence for a general slowing trend of tropical cyclones reported by Kossin (2018). We also suggest that the robust reduction in the tails of the acceleration distribution is indicative of a systematic change in the interaction of tropical cyclones with

extratropical baroclinic waves. We have not, however, examined the underlying processes. We speculate that poleward shift and decreasing amplitude of extratropical Rossby waves found in other studies may account for the acceleration trends. Detailed modeling and observational studies are needed to better understand the source of these trends.

## 11   Data and code availability

The reanalysis data used here can be obtained from the European Center for Medium-Range Weather Forecasts archived

at https://www.ecmwf.int/en/forecasts/datasets/reanalysis-datasets/era-interim/. The tracks of Atlantic tropical cyclones were obtained from the IBTrACS database available from https://www.ncdc.noaa.gov/ibtracs/. Authors will provide computer code developed for this paper to anyone who is interested.

## 12   Author contributions

AA wrote the computer code for all analysis and visualization, and wrote the text of the paper. TW derived the expression for

the radius of curvature, and assisted with editing the text and interpretation of the results.

## 13   Competing interests

The authors declare that they have no conflict of interest.

## 14   Financial Support

This work was supported by NSF through award #1433763



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
