# Peer review of "Acceleration of Tropical Cyclones As a Proxy For Extratropical Interactions: Synoptic-Scale Patterns and Long-Term Trends"

_Weather and Climate Dynamics, 2021_

## Author Comment (AC1)

**Responses to reviewer comments**

Anantha Aiyyer[1] and Terrell Wade[1]

[1]Department of Marine, Earth and Atmospheric Sciences, North Carolina State University

**Correspondence:** A. Aiyyer (aaiyyer@ncsu.edu)

We are grateful to the reviewer for their time and effort in helping improve the content of the paper.

**1 Point by point response**

This is a well-written manuscript, and I find it enjoyable to read. I have no major comments except some suggestions. I hope these suggestions would be helpful for the authors to improve the presentation of their findings.

**Response:** Thank you for taking the time to read and provide comments.

Title: This paper covers both the acceleration and deceleration of tropical cyclone motion. It is not clear whether "acceleration" here indicates general changes of motion speed or just an increase of movement speed. It might be helpful to rephrase the title to avoid confusion. A clarification in the abstract and/or main text would also be helpful.

**Response:** The acceleration here refers to the change in speed of the TC (i.e., motion) and not the change in the sustained winds associated with the TC (i.e., intensity).

Line 2: "While... has not". This sentence does not appear essential here.

**Response:** You definitely have a point here. We kept it to emphasize the novelty of our work. The bulk of TC motion studies have focused on translation speed and tracks. To the best of our knowledge, this is the first attempt at examining the curvature and tangential accelerations of TCs and their trends.

Line 2-4: This sentence probably needs some clarification. What are the interested "synoptic-scale patterns"? Are these patterns related to tropical cyclones or extratropical wave packets? It might also be useful to describe "tangential and curvature components of their (??) acceleration" more clearly.

**Response:** Since this is the abstract, it is not practical to provide all the details. The key aspects of synoptic-scale features are mentioned in the following lines. The definitions of tangential and curvature acceleration are provided in the text.

Line 18: Chan (2005) and references therein provide useful information on this topic.

- Chan, J. C. (2005). The physics of tropical cyclone motion. Annu. Rev. Fluid Mech., 37, 99-128.

**Response:** Agreed. This paper has been referenced by us.

Line 51: Some recent modeling studies present evidence suggesting that changes in extratropical weather could affect TC motion. These studies could help to better motivate this theme of investigation.

- Zhang, G., T. Knutson, and S. Garner, 2019: Impacts of Extratropical Weather Perturbations on Tropical Cyclone Activity: Idealized Sensitivity Experiments with a Regional Atmoshperic Model. Geophys. Res. Lett., 46, 14052– 14062.
- Zhang, G., Murakami, H., Knutson, T.R., Mizuta, R. and Yoshida, K., 2020. Tropical cyclone motion in a changing climate. Science Advances, 6(17), p.eaaz7610.
- Hassanzadeh, P., Lee, C.Y., Nabizadeh, E., Camargo, S.J., Ma, D. and Yeung, L.Y., 2020. Effects of climate change on the movement of future landfalling Texas tropical cyclones. Nature communications, 11(1), pp.1-9.

**Response:**: Thanks! We have added these citations.

Line 86-87: "... remain independent of modeled products to characterize the storms". The classification of recent storms by forecasters is partly based on models. But as recognized later, reanalysis datasets have issues with tropical cyclones, so the IBTrACS is still a reasonable choice.

**Response:** Agreed

Line 100: It would be helpful to conceptually link tangential and curvature changes of storm motion to physical factors (e.g., steering flow).

**Response:** That is a good point and would be a useful follow up exercise. Some aspects of the steering flow can be gleaned from the composites of geopotential height shown in this paper.

Fig. 3: Is it possible to mark the mean vectors and/or tracks of storm motion? This will help to infer how environmental flow affects the storm motion. Also, HGT1000 can be below sea level due to the low central pressure of TCs. It might be more

intuitive to use sea-level pressure here. Finally, the values of the colorbar are hard to read because of the small font size.

**Response:** A sense of storm motion has been provided in the composites. Please see the "+" symbols in Figs. 4, 6 and 7. Also, Figs. 5 and 8 show the track of the composite TC.

Fig. 5: For the acceleration case, it might also be helpful to plot the trough anomalies to support the argument that "tropical cyclone merges with the extratropical stormtrack" (Line 219). After all, trough anomalies are commonly discussed in the context of extratropical storms.

**Response:** Agreed. Anomalies are provided in Figs. 4, 6 and 7.

Line 257: During vortex interactions, a merger usually happens between vorticity anomalies of the same sign. It is a little odd to discuss a merger of positive and negative HGT/vorticity anomalies.

**Response:** Thanks for this. We have edited this line to avoid the misinterpretation highlighted by you. We were discussing the merger (or lack thereof) of the cyclonic anomalies of the TC and the upstream trough. The edited line now reads:

*By Day+2, we do not observe a merger of the TC and the upstream trough that happens in the case of rapid tangential acceleration (Fig. 5a).*

Section 6 and 7: The discussion is supported by the same figure and probably can be consolidated into one section.

**Response:** You have a point, but we have kept them separate for ease of readability.

Section 8: The transition from the discussion of translation speed to the discussion of acceleration is abrupt around Line 320. If this discussion of translation speed is deemed important, it probably should be consolidated with the discussion of Fig. 15 and Table 5 to keep the logic flow smooth. Otherwise, the results between 301 and 319 could be briefly summarized.

**Response:** The last couple of lines of section 7 serve as a clear end to the discussion on translation speed and then the stage is set for the acceleration.

Line 474: A weakening of extratropical cyclone activity is also projected by CMIP models (e.g., Chang 2013).

Chang, E. K. (2013). CMIP5 projection of significant reduction in extratropical cyclone activity over North America. Journal of Climate, 26(24), 9903-9922.

95

**Response:** Thanks for this reference. It has been included now.

Line 459-461: This part probably can be moved to the beginning of Section 8 to better motivate the trend analysis.

100    **Response:** We tried your advice but reverted back to the original text as it flowed better. Nevertheless, thank you for this suggestion.

Citation: https://doi.org/10.5194/wcd-2021-4-CC1

---

## Author Comment (AC2)

We thank the reviewer for their careful reading of the paper, their comments and for their time. Our responses are provided below each comment.

**1 Reviewer 2**

In this study, IBTrACS and ERA-Interim data on NATL tropical cyclones from 1966-2019 are used to investigate the synoptic-scale patterns that are associated with different characteristics of TC acceleration. Three major synoptic-scale patterns are identified: Rapid tangential acceleration of TCs occurs in cases with a developing extratropical wave packet, resembling a development typically observed during ET. Rapid curvature acceleration of a TC is linked to a dominant anticyclone east of the TC, initiating its recurvature. For rapid tangential deceleration and small (near-zero) curvature acceleration, the synoptic pattern resembles a cyclone-anticyclone dipole. Further, a statistical assessment on the characteristics of TC acceleration and speed is conducted, using quantile regression approach. The statistical analysis reveals that the extremes in tangential acceleration (both rapid acceleration and deceleration), the maximum of curvature acceleration, as well as the translation speed of TCs have decreased somewhat (negative trend) over the past five decades. The most robust negative trend has been found for a 20-50°N band during August and September.

This manuscript focuses on the characteristics on TC motion/acceleration during the interaction of a TC with the midlatitude flow in the NATL. It thus could help to complement the findings from previous work, which put a focus on the characteristics of the midlatitude flow and its wave packets during the interaction. In this context, the manuscript compares the results of this study with the findings of Riboldi et al., who focused their investigation on the translation speed of the upstream trough. Overall, the manuscript is well organized and written in a clear manner, and fits well into the scope of WCD, as it investigates both the dynamics perspective of the interaction as well as climatological aspects. However, the comparison to the results in the context of the work by Riboldi et al. is, in its current version, not fully convincing to me (see major comment 1). Furthermore, some parts could benefit from some more clarity in the description/discussion, and in other parts, information on the approach that has been used is lacking. Once these comments has been addressed during the revision of the paper, it will make a very suitable contribution to WCD.

**Response** We appreciate the reviewer's effort and numerous constructive suggestions.

**1.1 Major comments:**

The discussion of the results in the context of the study by Riboldi et al (2019) needs revision. First, from the discussion in paragraph 258-267, it is not clear whether the reference is made to the ACCEL or DECEL scenario of Riboldi et al., this information (reference to DECEL) is only made in the Discussion (l.421). On the one hand, it sounds reasonable that the deceleration of a trough, during phase-locking, also manifests in a deceleration of the TC. However, I am not fully convinced yet by the reasoning and figures that are presented in this manuscript. The DECEL scenario of Riboldi et al. is conducive to baroclinic interaction and leads to a (strong) amplification of the downstream wave pattern (TC acts as a "wave maker", Keller et al. 2019). For such a synoptic configuration, I´d expect to see a stronger upstream trough in Fig. 7f-j directly upstream of the TC, as e.g. in Fig. 10 a, c, e of Riboldi et al. Further, when comparing Figs. 4 f-j and 7 f-j with the DECEL scenario in Fig. 10 of Riboldi et al., the position of the TC with respect to the ridge appears to be different. While in the DECEL scenario, the TC becomes positioned in the western part of the ridge during the development, ahead of the upstream trough, where it supports ridge amplification. In Figs. 4 and 7, the TC rather appears to be placed south to the center of the ridge. From the figures provided, its contribution to ridge building is not directly obvious. Please expand on the discussion of these findings in the context of the work by Riboldi et al. This could e.g. include a tracking of the upstream trough to demonstrate the phase locked configuration, additional analysis of the PV (or eddy kinetic energy budget) to analyse the contribution of the TC to the amplification of the ridge, or another suitable means.

**Response** You make excellent points. We have made several changes to our reference to Riboldi et al. (2019). To be clear, our main intent was to acknowledge that, to our best knowledge, no prior study has examined tangential and curvature accelerations of tropical cyclones and the attendant synoptic patterns. The Riboldi et al. (2019) study is the closest to ours in a narrow sense. The focus of our paper is quite different and we have made changes to the way we refer to Riboldi et al. (2019) in the revised text to clarify this. Your suggestions for additional work are quite appropriate, but given the scope and the current length of this paper, we respectfully defer it to a follow-up study in order to establish firmer connections between this paper and Riboldi et al. (2019).

I am a bit confused by the explanation of the rapid curvature acceleration marking the point of the recurvature of a TC. The explanation in ll. 234-241, on the one hand, sounds reasonable to me. On the other hand, it appears that the (composite) TC in the case of rapid tangential acceleration also undergoes recurvature, as it is e.g. obvious from Fig. 5, but also from Figs. 4 compared to Fig. 7. Please further expand on the differences between the cases of rapid tangential acceleration and strong curvature acceleration to make this point on recurvature clearer. In this context, it could also be of help to see the individual tracks of the TCs included in the composite (e.g. plotting the tracks of all cyclones in the composite after the shift of the grids has been performed).

**Response** You make a very good point. Indeed, many of the cases in the composites of rapid tangential acceleration also undergo recurvature. So, the two are not independent. In a composite (or statistical) sense, however, a distinction can be noted in the timing of the peak accelerations. This is evident most clearly in fig. 9b. We note that the mean curvature acceleration peaks right around the objectively determined recurvature point. On the other hand, the composite TC experiences increasing tangential acceleration for at least 24 hours after recurvature. The net effect is that the forward speed of the TC continues to rise after recurvature while undergoing ET.

ll. 218-224: Please add information on how the ridge of the extratropical wave packet has been tracked. It would also be good to add information on why the track of the ridge has been used here, instead of the upstream trough, which wraps around the TC, pointing also to a merging of the TC with the extratropical wave packet.

**Response** The ridge was tracked by objectively determining the peak geopotential value (local maximum). We have added this information in the revised version of the paper. Tracking the upstream trough would still lead to the same interpretation albeit from a slightly different visual perspective. We chose the ridge because of the emphasis placed in the past literature on the downstream ridge building related to ET.

**1.2 Minor comments:**

l. 94-96: The statement on improved reliability of TC data during the satellite era is partly a repetition of paragraph ll.64-70. Consider revising.

**Response** We have removed this line.

l.140-141: Could you add a statement on what this implies?

**Response** This is just an observation of the statistical distribution of the two types of accelerations.

l.160: Centroid position of *all* storms?

**Response** No, just of the storms that went into the composites. That is, the subset based on the latitude ranges and thresholds used.

l.168: Given the distribution of curvature acceleration in Fig. 2, the 32 km/h per day as the "near-zero" curvature acceleration seems a bit high. Should this be 3 or 2, instead of 32?

**Response** Thanks for catching this typo. Indeed, the threshold for near-zero curvature acceleration was 3.3 km/hr instead of 32 km/hr. Also, note that the rapid curvature acceleration cut-off was 46 km/hr and not 48 km/hr. These have been corrected in the revised paper.

l.170: For completeness, please also state how many unique storms fall into the category of rapid and slow curvature acceleration.

**Response** We have now added the text to answer this: "These correspond to 196 and 168 unique storms for rapid tangential acceleration and deceleration respectively, and 170 and 149 unique storms for rapid and near-zero curvature acceleration respectively."

ll.177-181: This information is already contained in the figure caption (same applies to l 200-202). Consider revising.

**Response** Done. We have deleted the redundant sentences.

l.199 and later: The use of the word „system" to refer to the synoptic structure was a bit confusing to me, as system is typically also used to refer to e.g. a tropical cyclone. Consider revising throughout the manuscript

**Response** Done. We have replaced "system" with storm whenever it was used in the context of a TC. We use this term now only when describing the entire synoptic field (including the TC and the extratropical wavepacket).

l.210: Consider adding also information on day +2, as it is included in Fig. 4.

**Response** Day+2 is a mere continuation of what happens on Day+1 and we were hoping that the figure is self-evident.

l.211: The downstream trough has not been discussed before. Consider mentioning to already during the discussion of the wave packet in the paragraph above.

**Response** We have corrected this statement to read: "..downstream ridge-upstream trough couplet.." We are referring to the upstream trough here, not downstream trough. We agree that our original sentence was open to misinterpretation and we hope this corrects that.

l.213: From the figures presented, the strengthening of the geopotential gradient north of the storm is rather hard to detect (a tightening of the geopotential height isobars can somehow be identified in both the left and the right panels).

**Response** The panels are a tad cluttered owing to the space constraint but when viewed on screen (esp. with a little bit of zooming in), the increase in gradient can be seen, particularly poleward and eastward of the composite storm. We have added the word "eastward" to narrow down the region.

ll.215-216: Downstream dispersion of energy may occur in both cases (if the TC interacts with an existing wave packet, as well as if it excites a new one)

**Response** Agreed. That is the intent of this statement.

l.239: Consider adding a bit more information on the study/findings of Aiyyer (2015) here.

**Response** That paper was concerned with predictability of the downstream response related to recurving typhoons. We cite that paper here to provide a reference for the objective method of determining the recurvature point.

l.307: The "all storms" in Fig. 10 are shown in pink/magenta, but text and figure caption state grey.

**Response** Thanks for catching this typo. Fixed it.

l.309 and others: Thiel-Sen should read Theil-Sen **Response** Thanks for catching this typo. Fixed it.

l.309: According to figure caption, it should read 20-50°N band.
**Response** Thanks for catching this typo. Fixed it.

l.320: This section could benefit with a brief introductory sentence on its aim (even if it is just a sub-section), as e.g. the start into sections 6, 7 and 8, or like the sentences in L. 329-331. Consider adding.

**Response** Done. We now start this section with the following sentence: "We now examine trends in the accelerations. We begin with a overview of the shift in acceleration distribution over the past 5 decades."

l.325: bottom row -> There is just one row in Figure 12

**Response** Thanks for catching this typo. Fixed it.

l.325-327: The shifts in the CDFs for curvature acceleration are there, but not as clear as for the tangential acceleration, e.g. for 0-20N the 1988-1997 period appears to be characterized by more rapid acceleration that the prior and later period.

**Response** Agreed. We show the CDFs for a big picture view but rely on the Quantile regressions for a clearer quantitative estimate of the trends.

l.326: Consider adding "not shown" already after the sentence on the CDF over the entire year.
**Response** Done.

l.327: Could you comment on what might cause this increase in shifts seen in the CDFs, when October and November are omitted. October typically shows the highest percentage of TCs undergoing ET in NATL, e.g. Hart and Evans 2001.

**Response** It is unclear from our analysis. This will entail looking at the shifts in the storm tracks month by month, something that we are hoping to follow up in a later study

l.360: Could you comment on why the restriction has been made to August-September here, instead of e.g. September-October (same reason as above).

**Response** We chose Aug-Sep because these two are the busiest months of the Atlantic Hurricane season. These subsets are provided to illustrate some sensitivity to the choice of months.

l.362: I do not understand the reference to Table 1 (tau=.5) here. Do you refer to the 0.68 median tangential acceleration in Table 1 for "Full basin"? Please clarify. **Response** yes, that is what we mean here.

l.382: I am a bit confused by the statement that the OLS estimate of the trend is nearly the same value as it was for the annual-mean speeds. From Table 5, the OLS trend for the entire Atlantic and all months is -0.01, but from Table 2, for full basin and all storms, we get a trend of 0.029 (LR) and 0.028 (MK-TS), but maybe I am comparing the wrong information.

190   Consider adding a more specific comparison (e.g. number) for clarity.

**Response** Thanks for asking this. The comparison is being made with the numbers for the full basin (and excluding ET/NR). Those trends for speed are -0.007 (LR) and -0.008 (MK-TS). They are close to -0.01 value shown in Table 5. We have edited the sentence to read:

195   "For completeness, we also show the corresponding QR results for translation speed (Fig. 15 and Table 5). The OLS estimate of the trend for the entire basin ($-0.01$ km hr$^{-1}$ year$^{-1}$) is close to the trend calculated from the annual-mean speeds shown in Table 2 ($-0.007, -0.008$ km hr$^{-1}$ year$^{-1}$)"

ll.405-408: Please be more specific here on how the impact of phasing is evident in the rapid curvature acceleration (as you did above for the rapid tangential acceleration). The aspect of phasing is not discussed in the section 5.2.2.

**Response** Perhaps not with the same words, but we did discuss the arrangement of the tropical cyclone and the extratropical wavepacket earlier.

205   l.409: "for rapid tangential deceleration and near-zero curvature acceleration", as there is no curvature deceleration. Same applies to l.421 (rapidly decelerating TCs) and other instances. Consider revising throughout the manuscript.

**Response** Done. All instances have been replaced.

210   ll.476-477: For the negative trend in in translation speed and in curvature acceleration this statement sounds convincing, as well as for the negative trend in rapid tangential acceleration. However, could it also serve as an explanation for the observed decrease in rapid tangential deceleration?

**Response** Yes, the magnitude of rapid tangential deceleration has also decreased.

215

l.481: three (?) broad sets of synoptic-scale patterns

**Response** Yes, Corrected.

**1.3   Figures & Tables:**

220   Fig. 11 and 12: Labels are hard to read, consider enhancing their size.

**Response** Agreed.

Tab. 4: The OLS 95% confidence bounds are put in brackets here, but not so in Tab. 3 and 5. Consider harmonizing.

225

**Response** Agreed. Done

**1.4   Typos:**

11 T->t

230

**Response** Corrected.

48 shown->show

235    **Response** Corrected.

169: There is a bracket missing.

**Response** Corrected.

240

250: There is a superfluous space in the bracket for Fig. 7.

**Response** Corrected.

245    Corrected
258: 8a->8b

**Response** Corrected.

250    259: . . . poleward, the(?) tropical

**Response** Corrected to read: "The key point here is that the tropical cyclone-ridge system acts like a vortex dipole and is nearly stationary in the zonal direction."

255    Several instances: To my knowledge, it is more common to use "storm track" instead of "stormtrack"

**Response** Corrected.

---

## Author Comment (AC3)

We thank the reviewer 1 for their careful reading of the paper, their comments and for their time. Our responses are provided below each comment

**1 Summary and overall recommendation:**

The proposed paper has two main aims: first, characterizing the large-scale flow patterns behind rapid acceleration and deceleration of tropical cyclones outside the deep Tropics; second, revisiting the issue of the variability and trends in tropical cyclone (TC) motion from the perspective of acceleration and deceleration. The paper is written clearly and the exposition of the results is generally easy to follow. There are, however, a few conceptual misunderstandings in the interpretation of the results, in particular about the role of phase-locking and blocking during TC deceleration/acceleration and, more in general, during extratropical transition (ET). Other important issues concern the choice of subsets for the composite analysis and the significance of the trend analysis. Some careful revisions to make the scope and the results of the paper more precise are due before this research paper can be granted publication in Weather and Climate Dynamics.

**Response**: We appreciate the evaluation and look forward to clearing up any misunderstands on our part.

**2 Main comments**

■ The advantages of employing TC acceleration as a metric are not immediately obvious: the authors could consider elaborating on this aspect to make the motivation of this work stand out more clearly and, more in general, to contextualize its relevance for our understanding of ET and of its impacts. From the title, for instance, acceleration is meant to be used as a "proxy" for interactions between TCs and the extratropical circulation, but what is actually gained with respect to a similar analysis performed on, e.g., TC translation speed, by comparing subsets of rapidly moving and stagnating TCs on the 30°N-40°N latitude band? The same critique could be extended to the second part: what is the benefit of using the acceleration framework with respect to the simple translation speed? Wouldn't it be just a more convoluted way to re-obtain the results of Kossin (2018) and Lanzante (2019), while being still heavily affected by the limited length of the data record?

**Response:** Acceleration of tropical cyclones is commonly observed during interactions with extratropical cyclones. This is why we use it as a proxy for such interactions. Note the speed of a TC is a function of how long a storm has been accelerating or decelerating. While we could certainly recast our analysis using fast and slow moving storms, it will be complementary but not analogous. Another reason to choose acceleration is that past studies have shown that ET acts like an impulsive forcing (e.g., Torn and Hakim 2015; Aiyyer 2015) and that would be better captured by acceleration as opposed to speed. Kossin (2018) and Lazante (2019) were not concerned with extratropical interactions and we argue that using rapid acceleration as a metric allows us to do so in an intuitive and simple fashion. To our best knowledge, no prior study has examined tangential and curvature accelerations of tropical cyclones and the attendant synoptic patterns and we feel that this makes our study somewhat novel.

■ The authors should carefully consider their use of the concept of phase-locking, which appears to be different from the one established in the literature by Hoskins et al. (1985) and later employed, among others, by Riboldi et al. (2019) [R19]. In that context, phase-locking (or "phasing") represented the optimal flow configuration for enhanced and sustained baroclinic growth of an extratropical cyclone and the correspondent amplification of a downstream Rossby wave packet. It occurs when an upper-level positive (potential) vorticity anomaly (i.e., a trough) is located a quarter of wavelength upstream with respect to a low-level positive temperature anomaly, leading to 1) sustained tropospheric ascent (by vorticity and temperature advection) in the region of the cyclone and 2) mutual intensification of the two anomalies via the anomalous flow field induced by the anomalies themselves. Phase-locking is inherently three-dimensional, as it

consists of an interplay between features at upper- and lower-levels. There are, however, a few points in the manuscript where a more "two-dimensional" concept of phase-locking is considered and this makes the comparison with previous literature problematic. For instance, lines 229-230, "The phasing of the tropical cyclone and the extratropical wavepacket as led to the formation of a cyclone-anticyclone vortex dipole" (that are, however, one to the north of the other); lines 265-267, "we have viewed this as a phase-lock between the ridge and the tropical cyclone" (this is confusing and conceptually incorrect, as phase-locking occurs at best between an upper-level ridge and a low-level anticyclone during anticyclogenesis), or lines 417-418. Also at lines 426-430 the mechanism that holds dipole blocking stationary is described, rather than the phase-locking dynamics during ET.

**Response:** The notion of phase locking is rather general with applications to various problems in engineering and science. One elegant application of that notion is that barotropic or baroclinic instability can be interpreted as mutually reinforcing, phase-locked Rossby waves (Bretherton 1966; Hoskins et al. 1985). Other examples of phase locking outside the context of instability include the ones involving gravity waves (e.g., Ruppert and Zhang 2019), equatorial Kelvin waves (e.g., Baranowski et al., 2016) and ENSO (An and Wang, 2000) to name a few. However, your point is well taken and we recognize the potential for misinterpretation. We have removed the lines 265-267 to avoid that confusion.

■ More in general, it seems to me that the outlined results relate only marginally to R19, despite it being probably the closest analogue in the literature. For instance, the DECEL subset by R19 features enhanced downstream flow amplification and atmospheric blocking activity, following a classic ET pathway of rapid TC poleward motion ahead of a stagnating upper-level trough. That subset would then correspond to the rapidly accelerating TCs of the current manuscript; however, the discussion of the results relates R19's DECEL subset with the subset of rapidly decelerating TCs (lines 225-231, 419-424, 436-439). It is also not always clear whether the analogy is drawn with Fig. 10a or Fig. 10b in R19. Another questionable point is the parallel drawn with Fig. 10 of R19, as no vortex dipole was observed or discussed by R19 (lines 260-265, 419-424; see also the next comment).

**Response:** We wanted to acknowledge Riboldi et al. (2019) as we felt that this paper was the closest to our approach. That said, Riboldi et al. (2019) is very detailed and focused on Rossby wave amplification and blocking. We agree that the relevance is marginal. We have removed the lines that are inconsistent with the interpretation of Riboldi et al. (2019). Specifically, we have removed lines 260-265; 419-424 of original version of the paper.

The lines in the introduction now read:

*Of note, however, are Riboldi et al. (2019) and Brannan and Chagnon (2020) that are somewhat relevant to this paper. The former study examined the interaction of accelerating and decelerating upper-level troughs and recurving western North Pacific typhoons. Their key findings are: (a) In the majority of cases, a recurving tropical cyclone is associated with a decelerating upper-level trough that remains upstream; (b) The upper-level trough appears to phase lock with the tropical cyclone; and (c) Recurvatures featuring such trough deceleration are frequently associated with downstream atmospheric blocking. Brannan and Chagnon (2020) found that the flow response was sensitive to the relative speed between the recurving tropical cyclone and the extratropical wavepacket. These studies have highlighted the importance of tropical cyclone motion for the outcome of the ensuing interactions.*

■ For rapidly decelerating TCs, the authors often say that the TC and the anticyclone become "phased" in a configuration of atmospheric blocking (a dipole block; e.g., lines 409-418). This occurrence does not seem realistic, as the horizontal scale and the dynamical characteristics of the large-scale blocking anticyclone are completely different from the ones of the tropical cyclone, and the latter is "enslaved" to follow the large-scale flow induced by the former.

**Response:** This is a very valid point. While we did not claim that this is an example of what is considered *canonical blocking*, we agree that it may come across as such. We have removed the reference to blocking in the abstract and conclusions. We retain a passing reference to it but qualify it as follows (section 9; discussion).

*The combined system remains relatively stationary compared to the progressive pattern for rapid acceleration. This arrangement qualitatively resembles a dipole block — an important mode of persistent anomalies in the atmosphere. The canonical dipole block is depicted as a vortex pair comprised of a warm anticyclone and a low-latitude cut-off cyclone. The dynamics of blocked flows are rich and the subject of a variety of theories that are far from settled. In the present case, the slowly propagating cyclone-anticyclone pair is likely an outcome of a fortuitous phasing of the tropical cyclone and the extratropical ridge.*

■ Related to the previous main point 3), R19 described how blocking occurs at the end of the Pacific storm track a few days after ET completion (see their Fig. 3) and did not mention blocking occurring at the same longitude of the transitioning TC, or with the transitioning TC being part of it. A direct impact of the TCs "injecting" low-PV air in the block could occur and therefore inflate the ridge (as speculated in lines 216-217), but this needs to be proved. A simpler interpretation of the composites in Fig. 5 or 7 would involve a pre-existent slow-moving anticyclone, maybe an atmospheric block, that decelerates the north-eastward progression of the TC because of 1) the presence of easterlies on the southern side of the block and 2) (more speculatively) the presence of an "inverse beta-effect", due to that large-scale anticyclone locally reversing the planetary vorticity gradient and effectively "pushing" the TC away from the anticyclone (cf. the selective absorption mechanism described by Yamazaki and Itoh 2009). The authors should check, using an appropriate diagnostic, whether the presence of blocking follows or precedes TC deceleration and therefore modify their interpretation of the results and the discussion section (e.g., lines 423-430) of the manuscript.

**Response:** As noted in the previous point, we feel that it is best not to pursue this line of argument since we cannot do full justice to the material on blocking without a detailed analysis in the vein of what you have described above. We have removed the references to the qualitative similarities to dipole blocking to avoid any misinterpretation.

■ There is evidence of a poleward trend in jet position and extratropical wave activity due to global warming, but there is also evidence of a poleward trend in TC genesis and track that can compensate this, as the region favorable to sustain TCs expands northward. This is one reason why it is speculated that ET storms reaching Europe will increase in frequency as global warming progresses (e.g., Haarsma, R. 2021, https://doi.org/10.1029/2020GL091483 and references therein). How would the authors comment on that?

**Response:** This is a good point and we are looking at trends in ET using our data set. It is definitely possible that ET frequency may increase despite the acceleration trends. As we point out in a later response below, not all interactions lead to ET, and the trend in ET may not necessarily follow the trend in interactions. There is some evidence for a poleward expansion of tropical cyclone tracks (Kossin et al. 2014) and this might offset the deceleration effect due to the concomitant poleward migration of the extratropical baroclinic eddies. These preceding lines have been added to the revised paper (section 9).

**2.1 Specific comments:**

Line 24: even though it may seem reasonable to suppose it, are there references discussing the impact of wind shear on tropical cyclone motion? A possible one might be Jones et al. (2000, https://doi.org/10.1002/qj.49712657008).

**Response**: Agreed and included.

Line 36: isn't also Bieli et al. (2019) a more recent reference to justify this high percentage of transitioning TCs in the North Atlantic?

**Response**: Sure, we have referred to Bieli et al. (2019) at other locations in the paper.

Lines 37-39: this is the only place in the literature review where TC acceleration is cited, it would also be a good place to motivate why it is worth investigating it (see also main comment 1).

**Response**: See our response to comment 1.

Lines 43-44: the work by R19 is definitely relevant, but another relevant reference not included in the literature review might be the recent work by Brannan and Chagnon (2020, https://doi.org/10.1175/MWR-D-19-0216.1) who also tried to investigate phasing during ET and focused on the North Atlantic.

**Response**: Thanks. Done.

Line 163: composites are built from TCs between 1980 and 2016 (line 144), but here is said that anomalies are calculated with respect to the 1980-2015 seasonal cycle, as for bootstrapping (line 172). Earlier (line 97) it was said that the considered period for composites would have been 1981-2016. Why these differences?

**Response**: Sorry about the typo. It should all read 1980–2016. We have fixed it.

Lines 166-170: storm-relative ensembles are built using data between 1980 and 2016 (line 144) and are based on data drawn from 196 and 168 unique storms (line 169). From a rapid check on Wikipedia (), "only" 555 tropical systems occurred over the Atlantic. This means that 196+168=364 tropical systems (65.6%, almost 2 out of 3) would then be either rapidly decelerating or rapidly accelerating storms. Aren't these numbers very high? This percentage seems far from the top and bottom 10% that should be selected to build the composites. Does this total need to be split between curvature and tangential acceleration? Please explain.

**Response**: You are right, around 555 tropical systems occurred over the Atlantic during this period. As we note in section 5, we get a total of 3515 track points within 30–40°N. 10% that is ≈ 350. This is the number of samples in each composite. The key point is that we consider the top and bottom 10th percentiles of events, not of storms. We get 196 unique storms for rapid acceleration and 168 for rapid deceleration. This means that around 35% of the storms experienced rapid acceleration and 30% of the storms experienced rapid deceleration at some point in their lifetime within 30-40 °N. These fractions are comparable with fractions of storms undergoing recurvature or ET. Also, we found that some storms end up falling within both categories meaning that there is a shift from rapid acceleration to deceleration or vice-versa. We have not smoothed the track data from IBTRaCs, nor have we attempted to impose any discontinuity checks. To avoid miscommunication, we have replaced "unique" with "individual" and the relevant lines in the revised paper are now:

*These correspond to 196 and 168 individual storms for rapid tangential acceleration and deceleration respectively. Note that some storms appear in both categories. We have not smoothed or imposed any discontinuity checks on the cyclone tracks taken from IBTRaCs. Similarly, we get 170 and 149 individual storms for rapid and near-zero curvature acceleration respectively, with some common storms in both categories.*

Lines 171-174: As the same TC can sit in the 30-40°N latitudinal band for several, consecutive 6-hourly time steps (lines 148-149), the composites are likely built by averaging together consecutive time steps with very similar large-scale flow configuration. For instance, 196 (168) accelerating (decelerating) TCs correspond to 352 time steps in each subset, so each TCs contributes to the composites with 1.8 (2.1) consecutive time steps ( 12 hours). This effect of serial correlation needs to be accounted for during bootstrapping, otherwise this might lead to a significance test that is too "easy" to pass. Instead of drawing 352 random time steps, an appropriate combination of dates should be selected so that a substantial fraction consists of couplets of consecutive 6-hourly time steps.

185

**Response**: This is a good point, something that we did consider. One way to reduce serial correlation is to narrow the latitude band for composites. We examined the sensitivity to using narrower bands. It reduces the sample size but our interpretations of the results are unchanged. So, we are confident about the big picture outcome. On a more philosophical level, there appears to be a general acknowledgement across fields regarding an over-reliance on significance testing (see for example the comment by Amrhein et al. (2019) in Nature). We feel that the results pass the synoptic smell-test and that should be an important con-

190 sideration.

Line 172: are TCs in the period of study selected only between July and October? If this is not the case, then why is the bootstrapping performed only by sampling dates in these months? A more appropriate sampling should take into account the climatological distributions of composite elements by selecting a random date in a time interval centered around the time of

195 each rapid acceleration/deceleration and by attributing it to a random year, as done, e.g., in R19.

**Response** Very few storms form in the Atlantic outside the months of July-October. This is why all our composites and bootstraps are based on July–October. Our anomalies are already defined relative to to a synoptic climatology and this takes

200 care of issues such as seasonality in the fields.

Lines 211-217: it would be great if the points discussed in this paragraph were also backed up by some more quantitative analysis. Besides showing in the composites the strengthening jet streak (e.g., with zonal wind anomalies), no metric of downstream impact is employed in the study and it is not clear whether the two subsets have significantly different downstream

205 impact on the flow evolution. The meridional flow index (Archambault et al. 2013), the eddy kinetic energy framework (Quinting and Jones 2016) or Rossby wave packet amplitude (R19) can be possible choices.

**Response** We considered adding a section about this but felt that it would be excessive for this paper. That is why we referred to the canonical response documented in the studies mentioned above (and also in the paper). However, the reviewer

210 does have an important point regarding the difference in the downstream impacts for the different acceleration categories. We are examining this as a separate paper to contain the the scope and length of this paper.

Lines 232-253: the composites for strong curvature deceleration are very similar to the ones of tangential acceleration. How would the results change by considering positive and negative values of curvature acceleration according to the local concavity

215 of the track? If the circle associated with the radius of curvature is to the right (left) of the track, a positive (negative) value can be given to curvature acceleration. Would it make sense?

**Response** We did consider separating the curvature accelerations based on the sign of the radius of curvature. We decided not to pursue that direction at this point to avoid excessive categories and also to preserve the sample size. We leave this to a

220 later study to avoid expanding the scope of the paper.

Furthermore, the upper and lower decile of curvature acceleration are remarkably similar (line 168, 48 and 32km/hr day-1) despite the large variability in the size of the curvature circles of Figure 1 and 32km/hr day-1 is the same value of the upper decile of tangential acceleration: maybe it is worth double-checking if the values written in the manuscript are correct.

225

**Response** Thanks for catching this typo. Indeed, the threshold for near-zero curvature acceleration was 3.3 km/hr instead of 32 km/hr. Also, note that the rapid curvature acceleration cut-off was 46 km/hr and not 48 km/hr. These have been corrected in the revised paper.

230 Lines 246-257: the wrapping of the anticyclonic anomaly around the TC during recurvature has not (to my knowledge) been discussed in previous literature. Couldn't the significantly weaker ridge be due simply to blurring of the composite with

increasing lead time? It would be helpful if the authors could elaborate on this aspect a bit more.

**Response** This wrapping ( clearly seen in anomaly fields) is not likely due to the blurring of the fields. We only show sta-
tistically significant anomalies and the wrapping, or rather elongation, appears to be continuous in time. It is also seen in the
composites of Aiyyer (2015) that were based on recurvature points of Western North Pacific Typhoons.

Lines 351-353: This approach to trend estimation is likely affected by the presence of serial correlation in the data. Was the
independence of the acceleration values for each quantile verified? How would the results change if trends of quantiles were
computed for each year?

**Response** Serial correlation is accounted for during the significance testing. QR is the appropriate way to test for trends in
the tails of the distribution instead of conditioning and aggregating the data annually using thresholds (e.g. quantiles computed
for each year)

Lines 387-395: the interpretation of the trends would suggest that the interaction TCs- midlatitude storm track is occurring
less often. How does this result relate with trends in the occurrence of ET, in this or other studies?

**Response** This is a good question. Bieli et al. (2019) did not find a trend in ET fraction in global reanalysis over any basin
with the exception of the South Indian ocean. The reduction in trends in acceleration should be viewed with caution in this
context. We deem rapid acceleration to be a signal of interaction of the TC and an extratropical wavepacket. However, not all
interactions lead to ET, and the trend in ET may not necessarily follow the trend in interactions. Nonetheless, these interactions
have a significant impact on local and downstream weather even if ET never happens. Given that we have limited reliable data
in other basins, the choice is to rely on global reanalysis fields (e.g.. Bieli et al. 2019) or ensembles of climate model simu-
lations such as Zhang et al. (2020). We believe that this question needs an in-depth analysis in a future study. In the revised
paper, we have added these lines:

*A few additional issues need further investigation. First, the trends in tails of the acceleration are not as prominent during
October as compared to the earlier summer months. We have not accounted for this observation but one potential factor may
be relevant. There is some evidence for a poleward expansion of tropical cyclone tracks Kossin et al. (2014) and this might
offset the effect of the concomitant poleward migration of the extratropical baroclinic eddies. Second, given the reduction in
the rapid accelerations documented here, it is natural to ask whether the frequency of ET has changed. Bielli et al. (2019a)
did not find a trend in ET fraction over the years 1979–2015 in global reanalysis data over any basin with the exception of
the South Indian ocean. A caveat in this regard is that not all tropical cyclone interactions with extratropical baroclinic waves
lead to ET and their trends might be connected in more subtle fashion. Again, given limited reliable data on global statistics of
ET, ensembles of climate model simulations such as the one described in Zhang et al. (2020) might shed some light on this issue.*

Lines 405-406: what is precisely meant in this context by "the impact of the phasing"? See also major point 2.

**Response** The phasing here refers to the relative location of the TC and the extratropical Ridge-trough couplet. As shown in
other studies, the outcome of the interaction is sensitive to the locations of these features (see discussion on bifurcation points
for instance). As noted in response to major point 2, we have avoided the use of "phase locking" for our results in the revised
paper.

Lines 440-445: this paragraph is rather general, how exactly do the results of this study highlight/confirm the presence of
bifurcation points during ET? The authors could consider removing this paragraph, as the Discussion part is already rather long.

**Response**: Agreed, these lines are not directly relevant to our main results. However, we felt it was important to acknowledge
other work that focused on the outcome of TC-extratropical interactions and the sensitivity of the outcome to the phasing and

280     existence of bifurcation points.

    Lines 445-458: parts of this paragraph are a repetition of the results in Section 6-7 and could be merged with them. This would emphasize the real topic of this discussion paragraph, the question "Is rapid tangential acceleration a sign of imminent extratropical transition?" The following lines provide additional data, but do not give a clear answer to this question, that is left
285 (unsatisfactorily) pending. A more direct answer about whether it is possible to employ acceleration as a proxy of ET could be helpful and might be emphasized in the paper (see also major point 1).

    **Response**: In the discussion section, we do point out that $\approx 65\%$ of the storms that comprised the composites for rapid acceleration completed ET within 3 days of the reference time transition as compared to $39\%$ for rapid deceleration. While not
290 a robust predictor of imminent ET, rapid acceleration does appear to favor it.

    Lines 469-470: the problem of "strength vs frequency" is very important for the interpretation of the results of the second, trend-related part of the manuscript. Without guidance on this aspect, the relevance of the outlined results is difficult to evaluate. The authors can provide some (at least partial) answers to this issue using the data in their possess and I encourage them
295 to try to do so. Trends in strength can be evaluated by considering the strongest/weakest acceleration in each season, trends in frequency by checking the number of TCs underdoing ET in each season or in the number of rapidly decelerating/accelerating TCs ($\tau<0.1$, $\tau>0.9$). The authors might likely have additional, better ideas.

    **Response**: Again, a very good point. Indeed, we have given this considerable thought and are following up on this aspect of
300 the study. We have already tried to define metrics for strength and frequency but are not convinced that we have a robust answer yet in part because we have data from just one basin and that too for a limited number of years. We need more fine grained data to get clearer answers regarding this issue. We feel that this kind of question is best answered via careful idealized numerical simulations where some of the dynamics can be controlled. While we can calculate statistics for ET and we are following up on that, the notion of interaction between the tropical and extratropical systems extends beyond ET and recurvature.
305

**2.2   Technical/style suggestions**

Line 25: the AMS Glossary of Meteorology refers to "storm track" rather than "stormtrack" (), please choose the more usual formulation (unless you have a strong argument to use "stormtrack").

310     **Response**: Done

    Lines 51-52: the sentence "While we begin. . ." can be omitted at this stage.

    **Response**: The sentence has been edited per suggestion. It now reads "We begin with translation speed to place our results
315 within the context of related recent work."

    Line 62: "to" natural factors, but "natural factors" is actually not very precise. Maybe just "attributed these discrete changes to regional climate variability as well. . ."

320     **Response**: Done

    Lines 75-76: what do the authors mean with "still" classified as tropical? Does it just mean that each storm must be classified as "TS" at least once in their life cycle to be considered?

325    **Response**: We have removed the word "still" to address the ambiguity. We only consider those instances of a given storm (i.e., speed and acceleration) when the storm was tropical in nature (i.e., it cannot be designated as ET or NR at that track point).

  **Response**: Done

330    Lines 92-93: just to be clear, only the time steps labeled as ET are omitted, right?

  **Response**: We omit time steps that are classified as ET, NR and MX (please see line 97 in the revised version of the paper or line 92 of the original version):

335    *Henceforth, we will collectively refer to the designations NR, MX and ET asnon-tropical, and unless explicitly stated, omit the associated track data in our calculations*

  Line 130: this "3-day threshold" was not introduced early.

340    **Response**: Please see line 79 of the revised paper (or line 74 of the original version).

  Lines 134-141: As many relevant data are in Table 1, Fig. 1 is not directly referenced in the discussion. Please consider to reference it, or omit it otherwise.

345    **Response**: You have a point. However, we prefer to keep it so that a visual description of the distributions as a function of latitude is also available to the reader.

  Lines 166-170: this paragraph can be merged with the bullet point ending on line 155, as it is its natural continuation.

350    **Response**: Done.

  Line 210: what is the characteristic signal of a cold front?

  **Response**: We were pointing to the elongated lobe/kink like structure seen in the 1000 hPa geopotential height contours.
355

  Lines 211-217: in which sense a poleward moving tropical cyclone may "either interact with an existing wave packet or [. . . ] perturb the extratropical flow"? The two options are not mutually exclusive. In terms of initiation of Rossby waves by TCs, Riboldi et al. (2018) is likely a relevant reference.

360    **Response**: Agreed. We did not mean to imply mutually exclusive options. The sentence has been amended to say: "The poleward moving tropical cyclone may either interact with an existing wavepacket and/or perturb the extratropical flow to excite a Rossby wavepacket that disperses energy downstream (e.g., Riemer and Jones 2014; Riboldi et al. 2018).
  Line 261 – PV was already introduced earlier.

365    **Response:** We have removed this abbreviation at all locations in favor of "potential vorticity"

  Line 278: sections 6 and 7 should be moved, in my opinion, in the rather short section 4 to characterize the evolution of tangential and curvature acceleration during ET. This would help introduce the acceleration framework for the rest of the study.

370    **Response:**

  Line 309: Theil-Sen

**Response:** Corrected all occurrences of this typo.

375

Lines 459-467: this paragraph seems a repetition of the results in the previous section, rather than a discussion item. It could be removed or merged with the description of the results in the previous section, or drastically shortened and attached to the following discussion paragraph.

380     **Response:** This paragraph has been shortened and combined with the following one.

---

## Author Response (AR2)

Thank you for bringing attention to the color scales. The revised paper uses color blind friendly colors and line types.